

# Soil stoichiometric characteristics and influencing factors in karst forests under micro-topography and microhabitat scales

Yi Dang[1,2], Hua Zhou[2,3], Wenjun Zhao[1,2], Yingchun Cui[1,2], Chengjiang Tan[4], Fangjun Ding[1,2], Yukun Wang[5], Run Liu[2], Peng Wu[1,2]

[1]Guizhou Libo Karst Forest Ecosystem Observation and Research Station, Guizhou Academy of Forestry, Guiyang, 550005, China

[2]Key Laboratory of National Forestry and Grassland Administration on Biodiversity Conservation in Karst Mountainous Areas of Southwestern China, Guizhou Academy of Forestry, Guiyang, 550005, China

[3]Guizhou Liping Rocky Desertification Ecosystem Observation and Research Station, Guizhou Academy of Forestry, Guiyang, 550005, China

[4]Maolan National Nature Reserve Administration, Libo, 558400, China

[5]School of Soil and Water Conservation, Beijing Forestry University, Beijing, 100083, China.

**Correspondence:** Run Liu (18285115229@163.com) and Peng Wu (zuishaoxu@163.com)

**Abstract.** To quantitatively evaluate the stoichiometric characteristics of karst forest soils and their response mechanisms to complex microenvironments, the study systematically investigated soil stoichiometric traits and influencing factors across micro-topography and microhabitat scales in the Maolan karst forest. Key findings include: (1) Soil nutrients (organic carbon, total nitrogen, hydrolyzable nitrogen, available phosphorus, available potassium, total calcium, exchangeable calcium, and exchangeable magnesium) exhibited strong variability with significant spatial heterogeneity; (2) Microhabitat factors significantly influenced nutrient accumulation, though different elements showed distinct response patterns to microhabitat variations; (3) Micro-topographic parameters (slope gradient, aspect, and position) exerted indirect effects through gravity, light exposure, and erosion, driving the formation of gradient patterns in soil stoichiometry; (4) Differential response mechanisms of nutrients to abiotic factors, combined with the differential nutrient regulation and absorption strategies of various plant life forms, collectively shaped the complex stoichiometric characteristics. Synergistic interactions were observed among microhabitat-micro-topography-plant life form factors, with geomorphological abiotic factors playing predominant roles at this scale. Although biotic factors like plant life forms showed relatively weaker direct influences, their regulatory effects were closely interrelated with microhabitat-topographic factors. This multi-dimensional feedback mechanism between biotic and abiotic factors reflects the complexity of nutrient cycling in karst ecosystems.

**Keywords.** Karst; Micro-topography; Microhabitat; Plant life form; Soil stoichiometry; Maolan National Nature Reserve

## 1 Introduction

Ecological stoichiometry, as a discipline investigating chemical element ratios in organisms and their interactions with environmental systems, represents an emerging interdisciplinary frontier bridging ecology and biogeochemistry. This field




fundamentally examines how elemental homeostasis regulates and influences core ecological processes (e.g., organismal growth, organic matter decomposition, and nutrient cycling), playing pivotal roles in deciphering coupling mechanisms between energy flow and elemental cycling within ecosystems(Chen et al., 2024). In soil ecosystems, stoichiometric characteristics serve not only as critical indicators for assessing soil nutrient availability, microbial metabolic activity, and organic matter decomposition rates, but also elucidate the coupling relationships between key soil nutrients in biogeochemical cycles and ecological processes(Joshi and Garkoti, 2023). This theoretical framework unifies element allocation patterns within the soil-plant-microbe continuum into a coherent dimension, providing a quantitative analytical framework to decipher ecosystem resource limitations, stability, and adaptability(Sardans et al., 2021).

As a central pillar of ecological stoichiometry, soil stoichiometry provides critical insights into nutrient use efficiency, environmental stress acclimation mechanisms, and soil-mediated ecosystem rehabilitation potential(Chen et al., 2022). Soil stoichiometry governs vegetation nutrient acquisition strategies through bidirectional feedback mechanisms: (1) Soil processes regulate plant-available nutrient supply by modulating litter decomposition, mineralization processes, and microbial metabolic activity; (2) Conversely, plants actively shape soil nutrient pools via species-specific litter inputs and root exudation patterns, establishing a homeostatic balance that sustains ecosystem nutrient cycling efficiency(Zhang et al., 2024). In highly heterogeneous ecosystems, spatial stoichiometric heterogeneity of soils may act as a cryptic driver of vegetation succession patterns. Thus, deciphering soil stoichiometric signatures and their driving mechanisms provides a critical pathway to unravel ecosystem functionality and advance vulnerability management strategies(An et al., 2019). Karst regions in China, covering approximately 1.3 million km² (13.5% of the nation's terrestrial area), are globally recognized for their pronounced ecological vulnerability and habitat heterogeneity(Wu et al., 2025). This region exhibits complex microhabitats shaped by high bedrock exposure rates, intense karst dissolution, and shallow fragmented soil layers. The emergence of microhabitat heterogeneity inherently correlates with undulating terrain and distinctive vegetation mosaics. This geomorphic configuration distinct from typical landscapes has reorganized spatial distribution patterns of soil moisture and nutrients in karst systems, resulting in island-like heterogeneity patterns. Such heterogeneity not only exacerbates decoupling of major soil nutrient cycles, but may also compel plant-soil systems to invoke stoichiometric homeostasis adjustments to acclimate to resource constraints(Zhang et al., 2022). The Maolan karst region (Guizhou, Southwest China), as a representative karst ecosystem, exhibits marked divergence in soil physicochemical properties, nutrient cycling regimes, and ecological functioning compared to non-karst systems, with pronounced spatial heterogeneity in soil stoichiometric signatures. At the regional scale, climatic regimes and lithological substrates serve as primary determinants of soil nutrient distribution patterns, whereas at landscape and finer scales, the key drivers shaping stoichiometric heterogeneity are closely associated with azonal factors including geomorphic configurations, habitat-specific traits, and vegetation assemblages. Although numerous studies have investigated the stoichiometric characteristics of soil carbon, nitrogen, and phosphorus and their driving mechanisms at global or regional scales(Feng et al., 2024), research on soil nutrient distribution patterns and their regulatory factors in karst regions—particularly at the hillslope scale—remains limited. Furthermore, the mechanistic understanding of how soil stoichiometric traits respond to biotic and abiotic factors is still underdeveloped, which



significantly constrains our comprehension of soil-vegetation coevolution mechanisms in karst ecosystems. Based on this, the present study investigates the Maolan karst region by systematically collecting soil samples across diverse microhabitats and microtopographic conditions, integrating vegetation surveys of plant life forms, and analyzing soil stoichiometric

characteristics to address the following scientific questions:

1) Spatial distribution patterns and heterogeneity of major nutrient contents and stoichiometric characteristics in karst soils;

2) Interaction networks among soil nutrient elements and the intrinsic regulatory mechanisms governing their stoichiometric balance;

3) Relative contributions and interactive effects of microhabitat types, microtopographic features, and vegetation life forms

on soil stoichiometric traits.

This study employs multi-scale and multi-factor integrated analysis to elucidate the formation mechanisms of karst soil stoichiometry, providing novel theoretical frameworks for understanding soil-vegetation co-adaptation mechanisms in karst ecosystems, while offering scientific underpinnings for ecological restoration and sustainable management practices in this region.

**2 Study area**

The study area is located in the Maolan National Nature Reserve, Guizhou Province, situated in the transitional slope zone between the Yunnan-Guizhou Plateau and the northern Guangxi hills. The geographic coordinates range from107°52'10"to 108°05'40"E and 25°09'20" to 25°20'50"N. The topography exhibits a distinct northwest-high, southeast-low pattern, with elevations ranging from 430.0 to 1078.6 m (predominantly 550-850 m)(Zhou et al., 2022). This region features a typical

mid-subtropical monsoon humid climate, with meteorological data indicating: an annual average temperature of 15.3°C (coldest month: 5.2°C; warmest month: 23.5°C), annual precipitation of 1,752.5 mm (concentrated in summer), annual relative humidity of 83%, annual sunshine duration of 1,272.8 h, frost-free period of 315 d, and total annual solar radiation of 63.29 kW·m⁻²(Wen and Jin, 2019).

The geological structure is predominantly composed of limestone and dolomite, forming a typical bare karst peak-cluster

depression system with bedrock exposure exceeding 80%. Soil resources exhibit pronounced spatial constraints, primarily distributed in rock fissures with shallow and discontinuous profiles. Chemically, soils are characterized by high calcium content, base cation enrichment, and elevated organic matter, classified mainly as black limestone soils. The vegetation comprises primary karst evergreen-deciduous broadleaved mixed forests, representing azonal vegetation, with a forest coverage rate of 87.4%. This ecosystem stands as the best-preserved and most representative karst forest ecosystem at the

same latitude in the Northern Hemisphere(Zhou et al., 2022).



## 3 Study methods

### 3.1 Plot setting

Given that the study area is a national nature reserve and features highly fragmented karst habitats, making completely random sampling impractical, this study primarily employed stratified random sampling for plot placement. Specifically, based on a systematic survey of existing permanent plots at the Libo Karst Forest Ecosystem Positioning Observation and Research Station, five habitat strata were predefined: microhabitat, life form, slope degree, slope aspect, and slope position. Within each combination unit (e.g., sharp slope + semi-shady slope + downslope + stony gully + evergreen tree), random grids were generated. After field reconnaissance verified usability and excluded unsuitable locations, replacement sites were manually identified within the same stratum. Ultimately, all sampling points were successfully established. During field investigations, high-precision handheld GPS devices were used to measure geographic coordinates (latitude and longitude) and elevation data for each sampling site, professional compass clinometers were employed to determine slope aspect and gradient parameters, and slope position and microhabitat information were recorded through on-site observations(Figure 1).

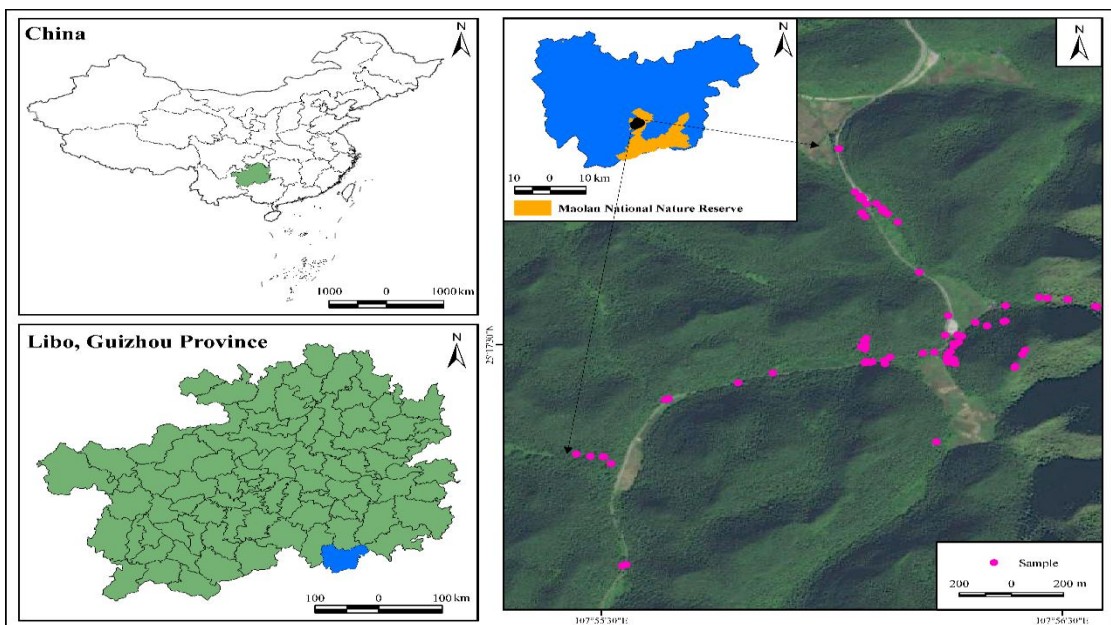

**Figure 1.** Spatial distribution maps of sampling sites in Maolan National Nature Reserve, Libo, Guizhou. Four-tier geolocation hierarchy: China's national framework (top-left) with green area indicating Guizhou Province; Libo County boundaries (bottom-left) with blue zone marking study townships; Maolan National Nature Reserve (top-right) with orange area delineating protected core zone; Satellite imagery of sampling sites (bottom-right) with pink circles designating soil sampling locations. (Scale bars: 1000 km/100 km/10 km/200 m; north arrows: N)



## 3.2 Microtopography division

The microtopographic features of the study area can be systematically classified according to the following scheme(Wu et al., 2025):

Slope position: upslope, midslope, downslope, and depression (four classes);

Slope degree: flat slope(≤5°), gentle slope (5−15°), tilted slope (15−25°), steep slope (25−35°), and sharp slope (≥35°) (five classes);

Slope aspect: shady slope (337.5−22.5°, 22.5−67.5°), semi-shady slope (67.5−112.5°, 292.5−337.5°), flat land, semi-sunny slope (112.5−157.5°, 247.5−292.5°), and sunny slope (157.5−247.5°) (five classes).

## 3.3 Microhabitat division

Microhabitat refers to the micro-scale environmental units where individual organisms or populations reside, characterized by local topography, substrate type, and microclimate. While no universally accepted definition exists regarding its spatial scale, this study adopts a karst forest microhabitat classification system established in prior research. Practically, microhabitats in the study area were categorized into three types—rock surface, rock groove, and soil surface—based on principles of representativeness, distinctiveness, and operational feasibility(Wu et al., 2025). The specific classification criteria are as follows:

The rock surface microhabitat refers to microenvironmental units where bedrock is directly exposed at the surface. Identification criteria require fulfillment of any of the following conditions: Bedrock exposure rate >50%, with solution groove depth >30 cm and surface soil cover area <1 m²; or development of shallow dissolution micro-landforms including shallow rock grooves, small rock troughs, shallow solution pits, and narrow rock crevices, all with vertical dimensions ≤30 cm; or rock surfaces with soil cover area <1 m² and thickness <20 cm.

The rock groove microhabitat refers to groove-shaped microenvironmental units formed through karst erosion processes. Identification criteria require fulfillment of any of the following conditions: bedrock exposure rate >50% with groove depth >30 cm and soil cover area <1 m²; groove depth <30 cm but soil layer thickness >20 cm; or bedrock exposure rate <50% but presence of soil-dominated grooves with depth >30 cm.

The soil surface microhabitat refers to microenvironmental units with continuous and homogeneous soil coverage and relatively well-developed soil horizons. Identification criteria require fulfillment of either of the following conditions: continuous soil cover with an area ≥1 m², or soil cover area <1 m² with bedrock exposure rate ≤50% and absence of grooves deeper than 30 cm.

## 3.4 Sample processing and determination

Due to the typical karst terrain of the study area, characterized by an extremely thin surface soil layer, samplers completed soil collection using only a shovel and cutting rings (100 cm³), without requiring tools like soil augers designed for deep soil



sampling. Upon reaching a sampling point, the area within a 20 cm radius of plant root distribution was selected. After removing surface litter, three intact soil cores were collected using cutting rings. Following the manual removal of non-soil materials, the samples were thoroughly mixed, ensuring a net weight of ≥500 g per sample. The samples were immediately coded, sealed in sterile sampling bags, and temporarily stored in a portable cooler for subsequent analysis.

Samples were processed in batches upon returning to the laboratory. For the determination of pH, hydrolyzable nitrogen (HN), available phosphorus (AP), available potassium (AK), exchangeable calcium (ExCa), and exchangeable magnesium (ExMg), the soil samples were passed through a 2-mm nylon sieve. This procedure preserved the intact soil structure, avoided the destruction of active components through grinding, and ensured that the extraction of readily available nutrients reflected true field conditions. For the determination of soil organic carbon (SOC), total nitrogen (TN), total phosphorus (TP), total potassium (TK), total calcium (TCa), and total magnesium (TMg), the soil samples were passed through a 0.149-mm sieve. This ensured complete sample homogenization, eliminated particle size effects interfering with total element analysis, and guaranteed complete digestion/fusion..

Soil sample preparation strictly adhered to the Chinese Forestry Industry Standard LY/T 1210-1999. Analytical protocols included: pH determination via potentiometric method; soil organic carbon (SOC) quantified by potassium dichromate oxidation-external heating; total nitrogen (TN) via semi-micro Kjeldahl digestion; hydrolyzable nitrogen (HN) by alkali-hydrolyzable diffusion; total phosphorus (TP) using alkaline fusion-molybdenum-antimony anti-spectrophotometry; available phosphorus (AP) extracted by HCl-$H_2SO_4$ solution; total potassium (TK) measured through alkaline fusion-flame photometry; available potassium (AK) extracted with ammonium acetate-flame photometry; total calcium (TCa) and total magnesium (TMg) determined by atomic absorption spectrophotometry (AAS); exchangeable calcium (ExCa) and magnesium (ExMg) analyzed via ammonium acetate exchange-AAS. All procedures rigorously followed the Forestry Industry Standard of the People's Republic of China (LY/T 1210–1275–1999)(Table 1)(Yang and Da, 2006).

**Table 1.** Comparison of Detection Methods and Core Instrumentation for Soil Indicators.

| Detection Indicator | Standard Method | Core Instrumentation |
|---|---|---|
| pH | Potentiometric method (soil-to-water ratio 2.5:1) | pH meter (accuracy ±0.01) |
| SOC | Potassium dichromate oxidation - external heating method | Oil bath (180°C ±0.5°C), Titration apparatus |
| TN | Semi-micro Kjeldahl method | Kjeldahl nitrogen analyzer, Digestion block/furnace |
| HN | Alkaline hydrolysis - diffusion method | Constant temperature incubator, Diffusion dish |
| TP | Alkali fusion - molybdenum antimony anti-spectrophotometry | Muffle furnace, Spectrophotometer |
| AP | HCl-$H_2SO_4$ extraction (Bray P-1 method) | Oscillator/shaker, Spectrophotometer |
| TK | Alkali fusion - flame photometry | Muffle furnace, Flame photometer |
| AK | Ammonium acetate extraction - flame photometry | Centrifuge, Flame photometer |
| TCa & TMg | Atomic absorption spectrophotometry (total content) | Atomic Absorption Spectrometer (AAS) |
| ExCa & ExMg | Ammonium acetate exchange - AAS method | Centrifuge, Atomic Absorption Spectrometer (AAS) |




### 3.5 Data processing and analysis

This study utilized mass content to characterize soil nutrient indices, with elemental stoichiometric relationships calculated as mass ratios. Data analysis was performed using SPSS 25.0 and Excel 2016 for statistical processing, and graphical outputs were generated via Origin 2021. The Kolmogorov-Smirnov (K-S) test was applied to assess data normality. Prior to

correlation analysis, raw data were logarithmically transformed [ln(x+1)] to meet ANOVA assumptions and normality requirements. Homogeneity of variance was tested before conducting ANOVA, with LSD or Tamhane's T2 methods selected for multiple comparisons based on test outcomes(Nagamatsu et al., 2003).

Redundancy analysis (RDA) in CANOCO 5.0 was employed to evaluate associations between soil stoichiometric characteristics and environmental factors. Environmental variables were numerically coded as follows: slope positions

(upslope: 1, midslope: 2, downslope: 3, depression: 4), slope degrees (flat slope: 1, gentle slope: 2, tilted slope: 3, steep slope: 4, sharp slope: 5), slope aspects (shady slope: 1, semi-shady slope: 2, flat land: 3, semi-sunny slope: 4, sunny slope: 5), microhabitats (soil surface: 1, stong gully: 2, stone surface: 3), and plant life forms (evergreen trees: 1, deciduous trees: 2, shrubs: 3, herbs: 4). Monte Carlo permutation tests were conducted to assess the significance of constrained ordination models and quantify the effects of individual environmental factors on soil stoichiometric characteristics. Variance

partitioning analysis (VPA) was conducted using the vegan package in R 4.4.1 to quantify the explanatory contributions of environmental factors and their interactions to soil stoichiometric characteristics, with model goodness-of-fit evaluated by adjusted $R^2$ values.

## 4 Results and Analysis

### 4.1 Stoichiometric characteristics of soil in the Maolan Karst region

#### 4.1.1 Statistical characteristics of soil nutrient contents

The Kolmogorov-Smirnov (K-S) test revealed that most soil nutrients in the study area deviated from normal distributions, except for total phosphorus (TP) and total potassium (TK). The arithmetic means of TP and TK were 0.87 g·kg$^{-1}$ and 13.95 g·kg$^{-1}$, respectively (Table 2). Maximum nutrient concentrations predominantly occurred in stong surface microhabitats and

downslope positions, while minima were typically associated with soil surface microhabitats and midslope positions.

In terms of data dispersion, only TP, TK, and TMg exhibited moderate variability, with TK showing the lowest dispersion (CV = 0.26). Other nutrients displayed strong variability, particularly AP, which demonstrated the highest variability (CV = 0.95).




**Table 2.** Characteristics of various types of nutrient content.

| Statistical characteristics | Soil Organic Carbon /(g·kg⁻¹) | Total Nitrogen /(g·kg⁻¹) | Hydrolyzable Nitrogen /(mg·kg⁻¹) | Total Phosphorus /(g·kg⁻¹) | Available Phosphorus /(mg·kg⁻¹) | Total Potassium /(g·kg⁻¹) | Available Potassium /(mg·kg⁻¹) | Total Calcium /(g·kg⁻¹) | Exchangeable Calcium /(cmol·kg⁻¹) | Total Magnesium /(g·kg⁻¹) | Exchangeable Magnesium /(cmol·kg⁻¹) |
|---|---|---|---|---|---|---|---|---|---|---|---|
| Arithmetic mean | 125.08 | 9.52 | 495.24 | 0.87 | 2.48 | 13.95 | 271.65 | 17.72 | 10.49 | 6.51 | 3.73 |
| Geometric mean | 103.10 | 11.20 | 575.80 | 0.99 | 3.88 | 14.45 | 322.83 | 22.91 | 12.43 | 7.14 | 4.28 |
| Median | 104.22 | 10.51 | 500.08 | 0.98 | 2.45 | 14.48 | 272.06 | 19.98 | 11.40 | 6.64 | 3.75 |
| Standard deviation | 78.45 | 6.21 | 292.87 | 0.47 | 3.71 | 3.79 | 191.09 | 16.44 | 8.63 | 3.16 | 2.44 |
| Kurtosis | 0.46 | -0.40 | -0.48 | -1.09 | 0.77 | 1.45 | 0.02 | 3.36 | 8.15 | 1.42 | 2.23 |
| Skewness | 1.04 | 0.70 | 0.46 | 0.14 | 1.32 | 0.50 | 0.86 | 1.50 | 2.58 | 1.13 | 1.51 |
| Minimum | 21.30 | 2.45 | 112.70 | 0.19 | 0.27 | 7.21 | 84.17 | 2.83 | 3.87 | 1.99 | 1.21 |
| Maximum | 374.73 | 26.38 | 1326.12 | 2.11 | 15.39 | 27.55 | 943.74 | 94.78 | 48.69 | 16.90 | 13.03 |
| $P_{(K,S)}$ | 0.00 | 0.01 | 0.00 | 0.06 | 0.00 | 0.06 | 0.00 | 0.00 | 0.00 | 0.00 | 0.00 |
| Coefficient of variation | 0.62 | 0.55 | 0.50 | 0.47 | 0.95 | 0.26 | 0.59 | 0.71 | 0.69 | 0.44 | 0.56 |

Notes: CV ≤ 0.20 for weak variability, 0.20 < CV < 0.50 for moderate variability, and CV ≥ 0.50 for strong variability;. Same as below.

### 4.1.2 Soil stoichiometric characteristics across different slope degrees

The major soil nutrients exhibited distinct distribution trends across the five slope degrees. Maximum values were generally distributed across all gradients except gentle slopes, with a relatively higher frequency on flat slopes. Minimum values were primarily dispersed across all gradient types except flat slopes, showing a relatively higher frequency on sharp slopes and steep slopes. For example, the mean values of TN, HN, TP, AK, and TCa were highest on flat slopes. Specifically, the mean AK on flat slopes reached 476.29 mg·kg⁻¹, which was 1.48 times that of the mean across all other slope degrees, while on

sharp slopes it was only 230.17 mg·kg⁻¹, representing merely 71.30% of the overall mean. A significant difference existed between these two values ($P < 0.05$). However, the maximum mean values for a few elements, such as SOC, TMg, and ExMg, also occurred on slopes like sharp slopes. For instance, the mean TMg on sharp slopes (8.58 g·kg⁻¹) was 1.51 times that on steep slopes (5.69 g·kg⁻¹), with a significant difference between them ($P < 0.05$). Regarding stoichiometric ratios, maximum values were predominantly concentrated on sharp slopes and steep slopes for all ratios except TCa:TMg.

Minimum values were mainly dispersed across all gradient types except flat slopes, with a relatively higher frequency on tilted slopes. For instance, the mean SOC:TP on sharp slopes reached 248.77, exceeding that of all other gradients, while on tilted slopes it was only 98.18, representing 68.66% of the overall mean. The difference between these two values was significant ($P < 0.05$). However, the distribution trend of TCa:TMg differed from the others. Its mean value was highest on flat slopes (4.24) and lowest on gentle slopes (2.76), differing by a factor of 1.54, yet no significant difference was observed

(Figure 2).





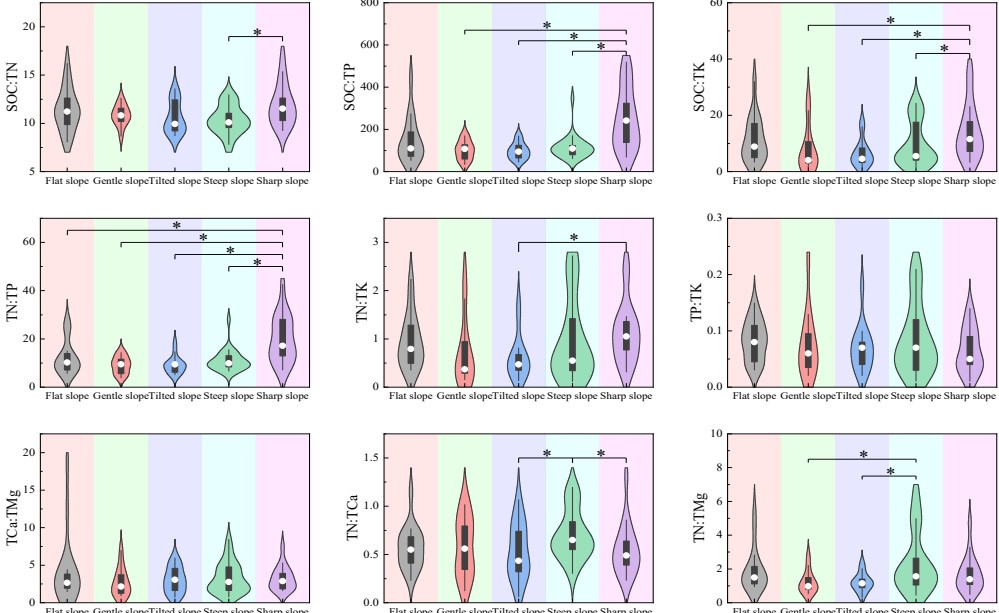

**Figure 2.** Distribution differences in stoichiometric ratios of major soil nutrients across different slope degree types, presented as violin plots overlaid with box plots. The Y-axis of each subplot denotes the values of corresponding ratios, while the X-axis represents slope degree types. An asterisk (*) indicates significant intergroup differences ($P < 0.05$), with black horizontal lines connecting groups exhibiting differences.

### 4.2.3 Soil stoichiometric characteristics across different slope aspects

Based on the results categorized by slope aspect, the major soil nutrients generally exhibited a distribution trend across the five slope aspects where flat land was the most enriched, followed by gradual depletion from shady slopes towards sunny slopes. Minimum values were almost exclusively concentrated on sunny slopes and semi-sunny slopes, with virtually no occurrence on flat land or aspects biased towards shade (shady slope, semi-shady slope). Maximum values were dispersed across all aspects except semi-sunny slopes, with a relatively higher frequency on flat land. For example, the mean AK on semi-sunny slopes was 223.71 mg·kg⁻¹, representing only 54.18% of the maximum value observed on flat land, and was significantly lower than all other slope aspects ($P < 0.05$). Additionally, the maximum value for individual elements also occurred on sunny slopes, such as TK reaching 15.90 g·kg⁻¹ on sunny slopes. However, due to heterogeneity of variances, Tamhane's T2 test revealed no significant differences among any of the slope aspects for TK. Regarding stoichiometric ratios of the major nutrients, minimum values were mainly concentrated on aspects like sunny slopes, while maximum values were more dispersed, showing no discernible distribution pattern. Constrained by the distribution of element contents across different slope aspects, most elemental stoichiometric ratios were generally similar among the aspects overall. Only a few ratios exhibited significant differences between specific aspect pairs, such as sunny slope vs. semi-shady slope, and sunny



slope vs. flat land, including SOC:TP, SOC:TK, and TN:TK. Specifically, the mean SOC:TP on sunny slopes was 97.93, representing only 68.48% of the mean across all slope aspects, and was significantly smaller than that on semi-shady slopes (164.50) ($P < 0.05$) (Figure 3).


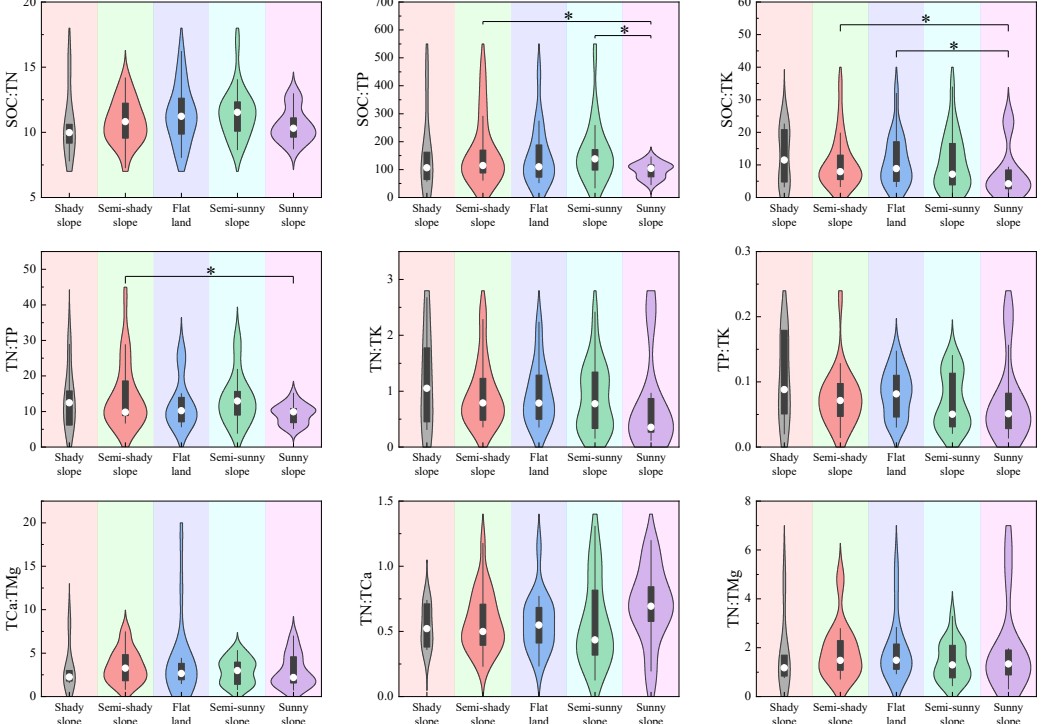

**Figure 3.** Distribution differences in stoichiometric ratios of major soil nutrients across different slope aspect types, presented as violin plots overlaid with box plots. The Y-axis of each subplot denotes the values of corresponding ratios, while the X-axis represents slope aspect types. An asterisk (*) indicates significant intergroup differences ($P < 0.05$), with black horizontal lines connecting groups exhibiting differences.


### 4.2.4 Soil stoichiometric characteristics across different slope positions

Maximum values of the major soil nutrients were distributed across all four slope positions, with a relatively higher frequency on the upslope (45.45%). Minimum values, although occurring on positions other than the downslope, were

primarily concentrated on the midslope (72.73%). Among the different slope positions, most elements generally exhibited the distribution trend: upslope > depression > downslope > midslope, such as TN, TCa, TMg, ExMg, and SOC. Under this distribution pattern, the maximum mean value for each element (on the upslope) often showed a significant difference compared to the minimum mean value (on the midslope) ($P < 0.05$). For example, the mean TN on the upslope was 13.59 g·kg⁻¹, which was close to its overall mean (11.20 g·kg⁻¹) but significantly higher than that on the midslope (5.98 g·kg⁻¹) ($P$

$< 0.05$), differing by a factor of 2.27. However, the distribution of stoichiometric ratios for the major nutrients varied





considerably among the slope positions. Maximum values occurred relatively frequently on the upslope, including SOC:TN, SOC:TP, SOC:TK, TN:TP, and TN:TK. This is primarily attributable to the higher contents of SOC and TN on the upslope. Minimum values mainly appeared on the midslope, primarily comprising ratios involving other elements with TK and TMg. This might be due to the relative enrichment of TK and TMg on the midslope, resulting in lower ratios. Based on the results

categorized by slope position, the stoichiometric ratios of various major nutrients were relatively similar between the downslope and depression, with almost no significant differences observed. Between the midslope and downslope, significant differences ($P < 0.05$) existed for almost all ratios except TN:TCa (Figure 4).

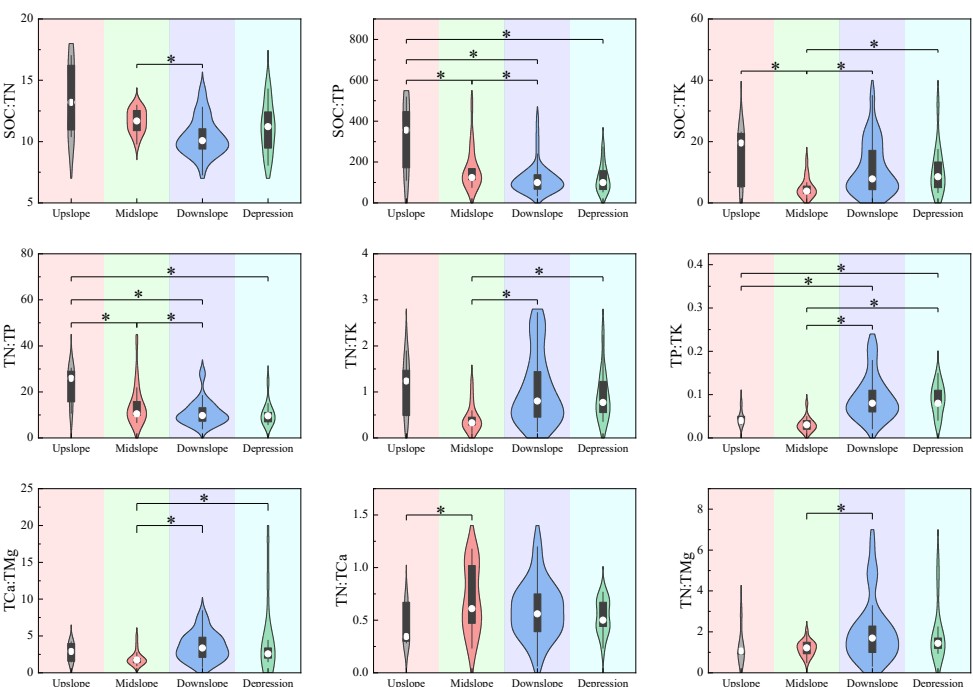

**Figure 4.** Distribution differences in stoichiometric ratios of major soil nutrients across different slope position types, presented as violin plots overlaid with box plots. The Y-axis of each subplot denotes the values of corresponding ratios, while the X-axis represents slope position types. An asterisk (*) indicates significant intergroup differences ($P < 0.05$), with black horizontal lines connecting groups exhibiting differences.

**4.2.5 Soil stoichiometric characteristics across different microhabitats**

The contents and stoichiometric ratios of major soil nutrients mostly exhibited similar trends across the three microhabitats. Specifically, the Soil surface microhabitat frequently exhibited minimum values, while the Stong surface microhabitat often exhibited maximum values, with significant differences commonly observed between these two. Values in the Stong gully microhabitat typically fell between these two. For example, the mean SOC content on the Stong surface reached 171.19





g·kg⁻¹, which was 1.37 times the overall mean across all microhabitats (125.08 g·kg⁻¹). In contrast, the mean on the Soil surface was 79.79 g·kg⁻¹, representing only 63.79% of the overall mean. The values between these two microhabitats differed by a factor of 2.15. This contributed to the trend where the mean SOC:TN in Stong gully (10.70) was less than that on Soil surface (10.70), which in turn was less than that on Stong surface (11.33). However, the distribution of TN:TCa was contrary to this overall trend. Its maximum mean occurred in the Soil surface microhabitat (0.66), while the minimum mean

occurred on the Stong surface (0.52), differing by a factor of 1.29, with a significant difference between them ($P < 0.05$) (Figure 5).

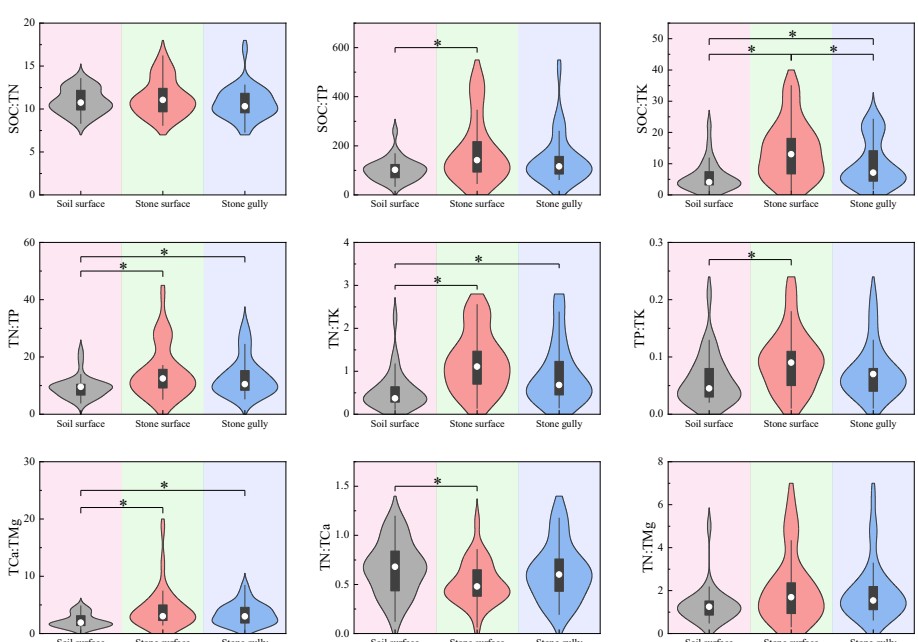

**Figure 5.** Distribution differences in stoichiometric ratios of major soil nutrients across different microhabitats, presented as violin plots
overlaid with box plots. The Y-axis of each subplot denotes the values of corresponding ratios, while the X-axis represents microhabitats.
An asterisk (*) indicates significant intergroup differences ($P < 0.05$), with black horizontal lines connecting groups exhibiting differences.

### 4.2.6 Soil stoichiometric characteristics across different life forms

Regarding the distribution of major nutrient contents and stoichiometric ratios in the root zones of the four life forms,
maximum values were predominantly concentrated in Herbs soil. Minimum values occurred across all four life forms but showed a relatively higher frequency in Evergreen trees or Deciduous trees soil. Among the different life forms, some elements conformed to the distribution trend: Herbs > Shrubs > Trees, with significant differences often observed between Herbs and Trees ($P < 0.05$). For example, the mean TN in Herbs soil was 14.99 g·kg⁻¹, showing significant differences compared to both Deciduous trees soil (9.34 g·kg⁻¹) and Evergreen trees soil (10.31 g·kg⁻¹) ($P < 0.05$). However, some





elements exhibited an opposite distribution trend. For instance, the mean TK in Deciduous trees soil (16.16 g·kg⁻¹) and

Evergreen trees soil (15.38 g·kg⁻¹) was significantly greater than in Shrubs soil (12.77 g·kg⁻¹) and Herbs soil (11.93 g·kg⁻¹).

This patterned distribution of element contents also resulted in certain distribution patterns for the major nutrient

stoichiometric ratios. Maximum ratios were frequently concentrated in Herbs (55.56%), while minimum ratios were more

distributed in Deciduous trees soil (44.44%). For example, stoichiometric ratios such as TN:TK and TP:TK were influenced

both by the aforementioned distribution trend of TK and by factors such as the TN and TP contents in Herbs soil being

greater than or significantly greater ($P < 0.05$) than those in Deciduous trees soil (Figure 6).

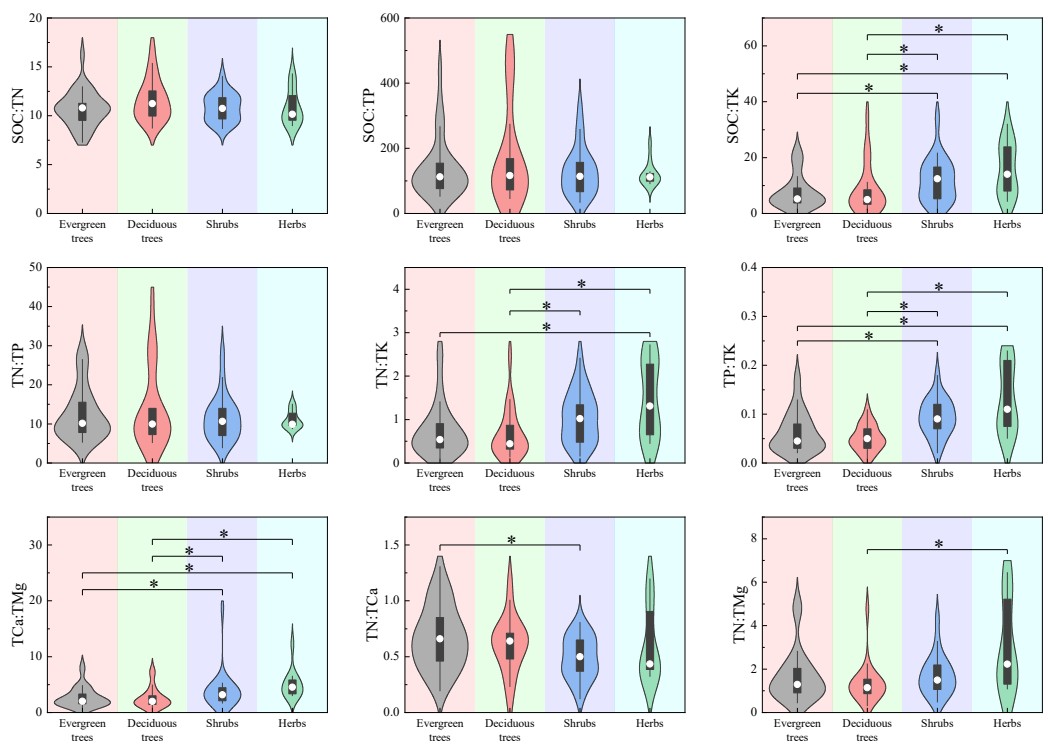

**Figure 6.** Distribution differences in stoichiometric ratios of major soil nutrients within the rhizosphere zones of different plant life forms,
presented as violin plots overlaid with box plots. The Y-axis of each subplot denotes the values of corresponding ratios, while the X-axis
represents plant life forms. An asterisk (*) indicates significant intergroup differences ($P < 0.05$), with black horizontal lines connecting
groups exhibiting differences.

**4.3 Correlation analysis between soil nutrient contents and stoichiometric ratios in karst regions**

The correlation matrix (Figure 7) revealed that SOC and TN contents exhibited significantly negative correlations with TK

($P < 0.01$), but significantly or highly significantly positive correlations with other major elements ($P < 0.05$ or $P < 0.01$).

AP showed no significant correlations with most elements except for significantly positive relationships with TCa and SOC.

For stoichiometric ratios, SOC:TP, SOC:TK, TN:TP, TN:TK, TP:TK, TCa:TMg, and TN:TMg demonstrated highly



significant negative correlations with TK ($P < 0.01$) but highly significant positive correlations with TN ($P < 0.01$). Overall,
elemental contents and stoichiometric ratios primarily displayed positive correlations, with negative correlations observed
between TP, TK, AP, TN:TCa, TN:TMg, and other parameters. Notably, when an elemental content shared the denominator
of a stoichiometric ratio (e.g., TP content vs. SOC:TP; TK content vs. TN:TK), correlations were predominantly negative,
some reaching high significance ($P < 0.01$). Conversely, when the element corresponded to the numerator (e.g., SOC content
vs. SOC:TN; TCa content vs. TCa:TMg), correlations were mostly positive, with several attaining high significance ($P < 0.01$).
$0.01$).

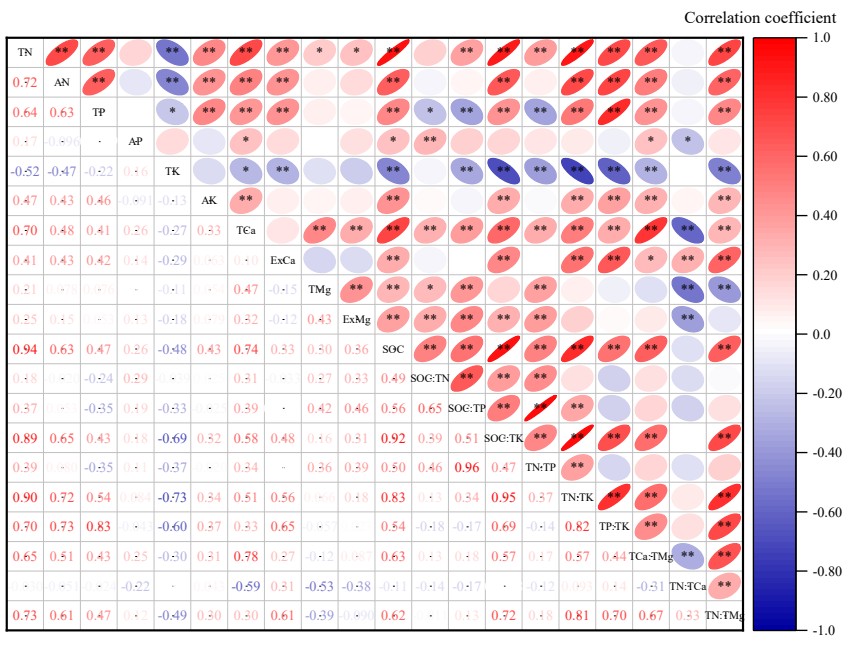

* $p < 0.05$    ** $p < 0.01$

**Figure 7.** Heatmap of Pearson correlation matrix for soil elemental contents and stoichiometric ratios. Ellipses in the lower triangular
matrix represent correlation coefficients (*r*): Color gradient indicates direction (red: positive correlation; blue: negative correlation);
Saturation and eccentricity indicate strength (proportional to |r|); Additionally, "*" denotes significant correlation ($P < 0.05$); "**" denotes
highly significant correlation ($P < 0.01$) Color scale ranges from -1.0 to 1.0 ("Correlation coefficient").

## 4.4 Influencing factors of soil stoichiometric characteristics in karst regions

### 4.4.1 RDA analysis

To investigate the effects of microtopography, microhabitat, and plant life forms on soil stoichiometric traits, redundancy
analysis (RDA; Figure 8) and correlation analysis (Table 4) were conducted. Results showed that the first and second RDA





axes explained 16.34% and 6.80% of the variance, respectively, cumulatively accounting for 23.14% of the relationships between soil nutrient contents/stochiometric ratios and environmental factors (microtopography, microhabitat, plant life forms). Monte Carlo permutation tests identified microhabitat as the strongest driver ($P < 0.01$), explaining 10.4% variance

with a contribution rate of 38.9%. Subsequent factors included slope position (8.0%), plant life forms (5.3%), and slope aspect (1.8%). Slope gradient exhibited no significant influence ($P = 0.15 > 0.05$), explaining minimal variance (1.4%) with a 5.0% contribution rate (Table 3). The RDA biplot (Figure 8) revealed positive correlations between microhabitat and TCa, TN, SOC, ExCa, TMg, HN, SOC:TK, TN:TK, TCa:TMg, TN:TP, SOC:TP, TP:TK, TN:TMg, while negative correlations occurred with TK and TN:TCa. Slope gradient positively correlated with AP, SOC:TP, TN:TP but negatively with AK, TP,

HN.

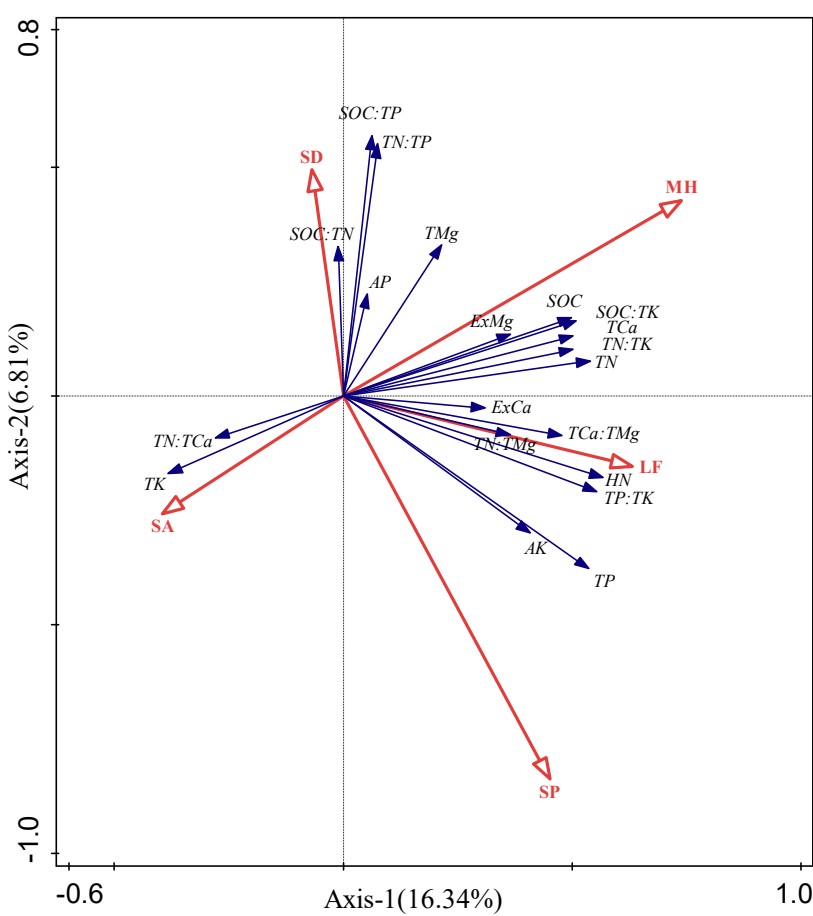

**Figure 8.** Ordination biplot of redundancy analysis (RDA) for soil stoichiometric traits and environmental factors. Axes: RDA1 (16.34% variance explained) and RDA2 (6.81%). Blue arrows: stoichiometric variables. Red arrows: environmental variables. Arrow length denotes

variable contribution; inter-arrow angles reflect correlations. Origin (0,0) serves as the reference point.




**Table 3.** Results of Monte Carlo test of effects of environmental factors on Soil stoichiometric characteristics.

| Environmental factor | Explains rate/% | Contribution rate/% | F value | P value |
|---|---|---|---|---|
| Microhabitat | 10.4 | 40.7 | 10.8 | 0.00 |
| Slope position | 8.0 | 27.1 | 7.8 | 0.00 |
| Life forms | 5.3 | 18.8 | 5.7 | 0.00 |
| Slope aspect | 1.8 | 7.9 | 2.4 | 0.02 |
| Slope degree | 1.4 | 5.6 | 1.7 | 0.09 |

**Table 4.** Correlation coefficients between soil stoichiometry characteristics and environmental factors.

| Stoichiometry | Correlation coefficient with each factors | | | | |
| | Microhabitat | Slope position | Slope aspect | Slope degree | Life forms |
|---|---|---|---|---|---|
| TN | 0.470** | 0.038 | -0.328** | 0.241* | 0.266* |
| HN | 0.296** | -0.221* | -0.169 | 0.386** | 0.451** |
| TP | 0.253* | -0.235* | -0.188 | 0.604** | 0.330** |
| AP | 0.259* | 0.301** | 0.031 | -0.215* | -0.027 |
| TK | -0.264* | -0.202 | 0.092 | 0.028 | -0.387** |
| AK | 0.245* | -0.451** | -0.283** | 0.522** | 0.113 |
| TCa | 0.508** | -0.02 | -0.306** | 0.19 | 0.317** |
| ExCa | 0.350** | 0.095 | -0.022 | 0.184 | 0.053 |
| TMg | 0.339** | -0.04 | -0.344** | -0.026 | -0.02 |
| ExMg | 0.216* | -0.019 | -0.440** | 0.18 | 0.366** |
| SOC | 0.456** | 0.039 | -0.306** | 0.142 | 0.258* |
| SOC:TN | 0.082 | 0.012 | 0.02 | -0.332** | 0.04 |
| SOC:TP | 0.283** | 0.266* | -0.18 | -0.416** | -0.03 |
| SOC:TK | 0.450** | 0.102 | -0.254* | 0.096 | 0.339** |
| TN:TP | 0.310** | 0.312** | -0.223* | -0.384** | -0.051 |
| TN:TK | 0.436** | 0.117 | -0.21 | 0.145 | 0.347** |
| TP:TK | 0.278** | -0.072 | -0.125 | 0.423** | 0.452** |
| TCa:TMg | 0.373** | -0.006 | -0.121 | 0.253* | 0.390** |
| TN:TCa | -0.227* | 0.067 | 0.152 | -0.031 | -0.184 |
| TN:TMg | 0.233* | 0.078 | -0.044 | 0.233* | 0.270* |


**4.4.2 VPA analysis**

To comprehensively assess the explanatory power of biotic and abiotic factors on soil stoichiometric characteristics, variance partitioning analysis (VPA; Figure 9) was conducted using microenvironmental factors (slope gradient, slope aspect, slope



position, microhabitat), plant structural factors (plant species, life forms), and plant nutrient factors (plant nutrient contents).

The results collectively explained 34.21% of soil stoichiometric variations. Microenvironmental factors (microtopography and microhabitat) alone explained 23.47% of the variation in soil stoichiometric characteristics, representing the highest individual contribution among all factor categories. In contrast, plant structural factors and plant nutrient factors individually accounted for only 2.5% and 1.04%, respectively, indicating minimal independent effects. Notably, interactions between microenvironmental and plant nutrient factors jointly explained 3.81% of the variation, while microenvironmental and plant

structural factors jointly explained 2.07%, both exceeding the interactive explanatory power of plant structural and nutrient factors (1.11%). However, the three-factor interaction exhibited the lowest explanatory capacity, jointly accounting for merely 0.22% of the variation.

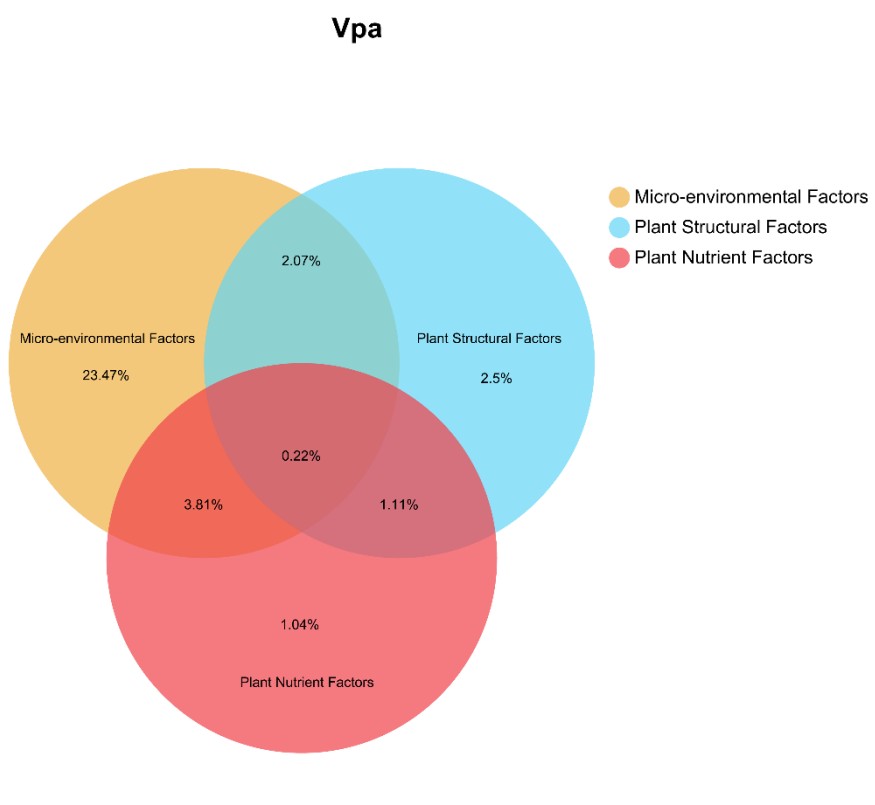

**Figure 9.** Variance partitioning analysis (VPA) of multi-factor contributions to soil stoichiometric traits. Tri-color overlapping system denotes environmental factor contributions: Yellow: Micro-environmental factors (slope/aspect/position/microhabitat); Blue: Plant structural factors (species/life form); Red: Plant nutrient factors (C/N/P contents); Values in overlapping areas indicate joint explanatory effects.



## 5 Discussion

### 5.1 Characteristics of Soil Nutrient Variation in Karst Regions

The soils in the Maolan Karst region primarily develop from carbonate rocks, characterized by slow pedogenesis, shallow soil layers, and extensive exposure of shallow bedrock(Tang et al., 2019). Complex fragmented landforms have formed due to soil erosion and karst processes(Peng and Dai, 2022). Under conditions where zonal climate and parent material are generally consistent, the formation of rocky desertification drives the differentiation and variation of soil nutrients through the regulation of surface environmental factors such as microtopography, small-scale habitats, and plant life forms(Zhang et al., 2018). These factors critically influence localized processes including solar radiation distribution, water movement, biological enrichment, and erosion disturbances. Consequently, they play a pivotal role in shaping heterogeneous patterns of soil stoichiometric characteristics at micro-scales(Waring et al., 2020).

Carbon is the most prevalent element in plant tissues(Prescott et al., 2020), while nutrients such as nitrogen (N) and phosphorus (P) serve as key limiting factors regulating plant growth and development(Krouk and Kiba, 2020). The study area exhibited relatively high overall contents of soil organic carbon (SOC, 125.08 g·kg⁻¹) and total nitrogen (TN, 11.20 g·kg⁻¹), surpassing the average levels found in karst forest soils of northwest Guangxi (92.00 g·kg⁻¹ SOC and 6.35 g·kg⁻¹ TN)(Zeng et al., 2015) and southern Yunnan's karst natural forest soils (31.36 g·kg⁻¹ SOC and 5.95 g·kg⁻¹ TN)(Huang et al., 2022). However, as shown by previous analyses, these two elements exhibited uneven spatial distribution patterns within the study area, primarily influenced by microhabitat factors, demonstrating strong spatial variability. The differences among microhabitats may reach significant or highly significant levels, which is consistent with findings reported in previous studies(Bi et al., 2024). Previous studies have revealed strong correlations between SOC, TN, and factors such as soil texture and vegetation growth(Zhong et al., 2021). These elements enhance the stability of soil nutrient retention capacity both directly and indirectly by promoting aggregate formation and stimulating microbial activity(Wang et al., 2019). Additionally, they improve soil nutrient input by regulating surface vegetation community structure, root growth, and litter dynamics, thereby establishing a positive feedback loop(Huang and Yuan, 2021). In this study, SOC and TN contents in stone surface microhabitats were significantly higher than those in soil surface microhabitats ($P < 0.05$), likely due to the uneven distribution and poor connectivity of organic matter in karst regions. Rock dissolution and soil erosion fragment the surface terrain, impeding nutrient migration. Soil eco-stoichiometric characteristics remained relatively stable within microhabitats(Wang et al., 2023). However, stone surface microhabitats exhibited enriched organic matter accumulation, higher macroaggregate content, greater mean weight diameter, and strong coupling between soil aggregate stability and carbon-nitrogen stoichiometric ratios. These properties suggest superior soil quality and elevated levels of SOC and TN in stone surface microhabitats(Bi et al., 2024). Similarly, TCa and ExCa displayed analogous distribution patterns strongly influenced by microhabitat factors. This reflects not only nutrient partitioning among microhabitats but also the aggregation effects of calcium(Toko et al., 2017). Studies have revealed that calcium in karst soils primarily originates from the weathering of parent bedrock and decomposition of plant and animal residues. Its release, accumulation, and migration are



influenced by soil development, microhabitat topography, and other factors(Luo et al., 2023). In stone surface microhabitats, where bedrock exposure rates are relatively high and connectivity is poor, external environmental factors such as rainfall and weathering accelerate calcium release from rocks, leading to elevated calcium levels in localized soils. In contrast, soil surface microhabitats exhibit weaker calcium enrichment effects and higher leaching losses due to greater distance from parent bedrock and stronger impacts of water leaching, resulting in significantly lower calcium content compared to stone surface microhabitats. Experimental studies have shown that under alkaline soil conditions, ExCa binds tightly to soil colloids. However, a decrease in soil pH promotes the exchange of ExCa with $H^+$ ions, causing its detachment from colloids into solution, where it becomes available for root uptake or susceptible to leaching loss. In this study, both pH and ExCa content were significantly higher in stone surface microhabitats than in soil surface microhabitats. This phenomenon may be attributed to the aforementioned mechanisms linking pH dynamics and calcium mobility(Han et al., 2019).

While most elements exhibited the highest concentrations in stone surface microhabitats, TK displayed a distinct distribution pattern, with significantly greater content in soil surface microhabitats than in stone surface microhabitats, primarily influenced by life form factors. Previous studies have indicated that soil TK content is predominantly controlled by macro-scale factors such as parent bedrock and climatic conditions, as different bedrocks vary in their potassium release capacities during weathering processes(Luo et al., 2014). In this study, parent materials and climatic conditions were nearly identical across sampling sites. However, arboreal species (e.g., trees) generally possess root systems with larger surface areas and greater distribution depths compared to shrubs and herbs. In soil surface microhabitats, where topsoil layers are relatively thicker, the expanded root-soil interfaces of arboreal species and associated microbial communities facilitate more active material exchange. This suggests that differences in root-soil interactions among vegetation life forms may be the primary driver of potassium distribution patterns(Bell et al., 2014). Although TK content reflects the soil's potential potassium supply capacity, vegetation primarily relies on AK for growth and development. In this study, AK content in downslope and depression areas was significantly higher than in upslope areas, likely linked to its mobility and transport(Jiang et al., 2016). Under rainfall and runoff erosion, AK in upslope and midslope positions may gradually leach and accumulate in downslope or depression areas.

In contrast to other elements primarily influenced by microhabitat and life form factors, phosphorus (P) and magnesium (Mg) were predominantly regulated by microtopographic factors. Soil total phosphorus (TP), representing the sum of various phosphorus forms, is determined by the combined effects of parent rock type, pedogenic processes, and organic matter input-output dynamics(Gao et al., 2021). In this study, the mean TP content reached 0.99 g·kg⁻¹, approximately 2.61 times the average level in subtropical evergreen broadleaf forests in China (0.38 g·kg⁻¹), with moderate overall variability. This limited variability may stem from the relatively uniform parent bedrock and climatic conditions across sampling sites. However, variations in solar radiation, evaporation, and microbial activity due to microtopographic factors (e.g., slope position and aspect) likely caused differences in litter decomposition and rainfall-driven organic matter fluxes(Wang et al., 2021). Previous studies suggest that high free calcium carbonate content in karst soils binds phosphorus into calcium phosphate salts, which are less available for plant uptake. This process may explain localized reductions in AP levels(Yang





et al., 2015). This study found AP content was most strongly influenced by slope degree, with significantly lower values in flat and gentle slopes compared to steep and sharp slopes. AP exhibited the highest variability among all nutrients (coefficient of variation, CV = 0.95), even reaching 1.23 in flat slopes. The gentle slopes in the study area are prone to seasonal waterlogging, while fragmented microtopography and complex microhabitats create heterogeneous free calcium carbonate distribution. These conditions drive strong spatial heterogeneity in calcium phosphate formation and AP availability. Similar to TP, TMg represents the summation of various magnesium forms in soils, including water-insoluble mineral-bound magnesium and ExMg that is adsorbed onto soil colloid surfaces and participates in cation exchange reactions(Takahashi, 2021). Both TMg and ExMg displayed pronounced slope aspect variability, with mean contents in shady slopes nearly double those in sunny slopes. This pattern may reflect magnesium's susceptibility to leaching(Jiang et al., 2021). The shady slopes in the study area supported dense shrub and herb vegetation, with a well-developed litter layer that played a critical role in enhancing soil stability and erosion resistance. These conditions thereby mitigated magnesium leaching loss caused by rainfall to some extent. Considering the distribution patterns of carbon, nitrogen, and phosphorus across different slope aspects, soils in shady slopes exhibited higher levels of major nutrient elements compared to sunny slopes. Consequently, enhanced activity of vegetation roots and soil microbial communities likely intensified elemental migration effects in these areas.

**5.2 Soil Stoichiometric Characteristics in Karst Regions**

Soil stoichiometric ratios, defined as the ratios between total elemental contents, are critical indicators for assessing the balance of soil nutrient supply potential and evaluating soil quality(Bai et al., 2020). The SOC:TN serves as a key parameter for understanding organic matter decomposition rates and mineralization dynamics, reflecting the carbon-nitrogen nutritional equilibrium in soils(Wang et al., 2018). Previous studies have demonstrated that when SOC:TN is below 16, enhanced microbial activity may drive decomposition rates exceeding organic matter accumulation. Conversely, SOC:TN values above 25 typically indicate net organic matter accumulation(Wang et al., 2016). In this study, soil C:N ratios ranged from 7.25 to 17.05, with a mean value of 10.83. This falls within the national average range for Chinese soils (10.00–12.00)(Zheng and Long, 2020) but is lower than the averages for subtropical forest soils in China (14.88)(Qiao et al., 2020) and the global mean (13.33)(Hessen et al., 2004). These findings suggest widespread microbial activity across most study areas, accompanied by relatively rapid decomposition and mineralization rates. However, SOC:TN exhibited weak overall variability, indicating relatively stable patterns. This stability may arise from synchronized responses of SOC and TN release and migration to environmental conditions. The strong correlation between SOC and TN implies that microbial decomposition of nutrient elements disrupts organic carbon skeletons while releasing nitrogen, favoring the maintenance of fixed stoichiometric ratios(Feng et al., 2021). This aligns with findings from other studies reporting stable SOC:TN ratios in soils(Li et al., 2017). The SOC:TP primarily reflects phosphorus mineralization capacity and is widely used to assess the potential of microbes to release phosphorus through environmental uptake or organic matter mineralization. Previous studies suggest that SOC:TP values exceeding 200 indicate carbon sufficiency coupled with phosphorus limitation, where microbial





competition for limited phosphorus resources may accelerate soil phosphorus depletion. Conversely, SOC:TP values below
200 signify net organic phosphorus mineralization, leading to increased soil phosphorus availability(Qin et al., 2016). In this
study, SOC:TP values were below the 200 threshold across all sampling sites except for upslope areas, which exhibited a
mean of 326.58. This supports the earlier conclusion regarding relative phosphorus scarcity in upslope soils. Integrated
analyses revealed that key stoichiometric ratios (e.g., SOC:TN, SOC:TP) were predominantly influenced by slope position,
with significantly higher values in upslope areas compared to downslope and depression areas. This pattern may be linked to
slope-specific plant nutrient allocation strategies. Studies have found that vegetation in karst upslope areas frequently
experiences nutrient loss due to runoff driven by microtopographic factors. Consequently, plants often adopt conservative
defensive growth strategies, prioritizing carbon accumulation to enhance stress resistance. Thus, the decomposition of plant
litter in upslope areas contributes to elevated SOC:TN and SOC:TP ratios by facilitating nutrient return to the soil(Dou et al.,
2024). Although the SOC:TN and SOC:TP ratios in upslope areas of the study region were greater than or significantly
greater than those in downslope or depression areas—indicating lower nitrogen (N) and phosphorus (P) contents in upslope
soils—phosphorus scarcity was more pronounced compared to nitrogen deficiency, as further evidenced by the TN:TP ratio.
This study revealed that nitrogen was retained more stably in soils than phosphorus with increasing slope position. This
disparity arises because phosphorus is less readily adsorbed by soil particles and is more susceptible to leaching losses
during erosive processes such as raindrop impact and runoff scouring, resulting in lower phosphorus availability in upslope
soils. The soil TN:TP ratio, widely recognized as a critical indicator of nitrogen-phosphorus equilibrium in previous studies,
reflects the soil's capacity to supply nutrients for plant growth. In this study, the TN:TP ratio in upslope areas reached 22.91,
far exceeding the average level in subtropical forest soils (12.43)(Qiao et al., 2020). This imbalance may further reduce
biological nitrogen fixation, leading to diminished levels of hydrolysable nitrogen (HN)—a directly plant-available nitrogen
form—in upslope areas(Yang et al., 2015), as evidenced by higher HN contents in downslope and depression areas
compared to upslope zones.

Given the relatively stable total potassium (TK) content in the study area—exhibiting moderate variability across both
microhabitats and life forms—fluctuations in the SOC:TK and TN:TK ratios were primarily driven by changes in SOC and
TN contents. The significant differences observed between microhabitats align with the distribution trends of SOC and
TN(Lei et al., 2024). In contrast, the TP:TK ratio appeared more susceptible to variations in TK. This may be attributed to
the relatively low TP content (0.19–2.11 g·kg$^{-1}$), which was substantially lower than the TK content range (7.21–27.55
g·kg$^{-1}$). Consequently, TK fluctuations exerted stronger control over the TP:TK ratio, leading to similar trends across
microhabitats as TK content varied(Wang et al., 2018).

Although most current research focuses on soil C:N:P:K(Qiao et al., 2020), key elements such as calcium (Ca) and
magnesium (Mg) also play critical roles in maintaining ecosystem functions(Liu et al., 2021). Their stoichiometric ratios
often influence the potential for vegetation to access these nutrients to some extent(Otero et al., 2013). Among these ratios,
TCa:TMg is commonly used to characterize the equilibrium between calcium and magnesium in soils and indicate soil
maturity(Han et al., 2021). This ratio is typically influenced by parent material composition, mineralogy, weathering





intensity, soil developmental stage, and vegetation type. This study found that soils under shrub and herb vegetation exhibited higher or significantly higher TCa:TMg ratios compared to arboreal species (trees), while TMg showed no significant differences across life forms. In absolute terms, TCa in the study area demonstrated stronger variability, with a mean content (17.72 g·kg⁻¹) nearly triple that of TMg (6.51 g·kg⁻¹). Thus, fluctuations in TCa exerted greater influence on the TCa:TMg ratio, consistent with prior correlation analyses. Previous studies suggest that vegetation growth may induce selective depletion of soil exchangeable alkaline-earth metals, significantly altering calcium-magnesium equilibria(Lu et al., 2014). Variations in calcium content driven by geochemical processes may relate to differences in calcium storage capacities among life forms. Specifically, arboreal species often exhibit stronger absorption and retention of calcium in rhizospheric soils under most conditions(Vittori Antisari et al., 2013). Furthermore, litter from arboreal species, once shed from the canopy, rarely remains stably aggregated around tree trunks. It is typically transported by wind, gravity, and runoff to accumulate and decompose within shrub-herb patches. Subsequent mineralization releases calcium into soil solutions, where it becomes partially available for reabsorption by fine lateral roots(Schmitt et al., 2013). This phenomenon aligns with findings from other studies. For instance, cedar litter with a mean calcium content of 17.9 g·kg⁻¹ was observed to release calcium into soils via leaching during decomposition, partially mitigating soil calcium loss(Takahashi, 2021). In this study, ground-dwelling shrub-herb vegetation outperformed canopy-dominant arboreal species in protecting and replenishing soil calcium through reduced rainfall impact, attenuated runoff erosion, and litter interception. This resulted in higher total calcium content and TCa:TMg ratios in shrub-herb microhabitats(Mao et al., 2022).

Although no significant differences in TMg were observed among soils under different plant life forms, we found that herbaceous soils generally exhibited lower TMg contents, with means ranging from 78.06% to 83.78% of those in other life forms. In contrast, TN in herbaceous soils was 1.22–1.60 times higher than in other life forms, resulting in herbaceous soil TN:TMg ratios that were greater than or significantly greater than those of other life forms. This pattern may relate to the negative correlations between soil nitrogen and magnesium reported in previous studies. Researchers have suggested that excessive nitrate leaching in nitrogen-rich forest soils may contribute to magnesium depletion(Wang et al., 2022). Conversely, relatively high magnesium levels can induce soil compaction, impairing permeability and aeration(Brock et al., 2021). Studies have indicated that whether magnesium supplementation promotes plant growth and fruiting depends largely on soil nitrogen availability(Grzebisz, 2013). Magnesium plays a critical role in root nitrogen uptake by enhancing nitrogen use efficiency and stimulating biological nitrogen fixation(Kiss et al., 2004). It also indirectly regulates nutrient acquisition by modulating plant-microbe symbiosis(Sun et al., 2018) and further facilitates photosynthesis and protein synthesis/transport(Farhat et al., 2016). Thus, the total nitrogen to total magnesium (TN:TMg) ratio may reflect the integrated capacity of plants to acquire and utilize major nutrients. In this study, herbaceous plants predominantly grew under forest canopies with heavy shading from arboreal and shrub vegetation. Consequently, their growth may be constrained by high TN:TMg ratios and limited light capture capacity.

Additionally, this study found that although both TN and TCa exhibited the highest mean contents in stone surface microhabitats and the lowest in soil surface microhabitats, the TN:TCa ratio displayed an entirely opposite distribution



pattern. This discrepancy may be linked to variations in calcium release across microhabitats. As revealed in prior research, soil calcium content is influenced by factors such as bedrock exposure rates. In stone surface microhabitats, limestone weathering-induced calcium release is typically more pronounced than in soil surface microhabitats due to higher rock exposure. Consequently, while stone surface microhabitats showed elevated TN and TCa contents compared to soil surface microhabitats, the increase in TCa exceeded that in TN, leading to a significant reduction in the TN:TCa ratio. Studies have highlighted calcium's critical role in facilitating the assimilation of other nutrients and stimulating microbial activity— functions essential for organic matter transformation processes(Lukina et al., 2019). In subtropical regions prone to soil acidification(Tian and Niu, 2015), intense leaching of alkaline cations such as $Ca^{2+}$ from TCa may force surface vegetation to rely on existing soil calcium reserves(Perakis et al., 2006). This implies that plants in soil surface microhabitats could face calcium limitation during later growth stages.

**5.3 Influencing Factors of Soil Stoichiometric Characteristics in Karst Regions**

Variations in soil stoichiometric characteristics in karst regions result from interactions between biotic and abiotic factors, reflecting the distribution and cycling patterns of soil nutrients across microhabitats. As indicated in prior studies, microtopography, microhabitats, and life forms collectively regulate soil nutrient input, migration, accumulation, and loss through complex interactions, serving as key drivers of nutrient cycling in karst forest ecosystems. Microtopographic factors influence spatial nutrient distribution by modulating localized slope runoff, light availability, temperature, and moisture conditions. For instance, slope gradient primarily affects water and nutrient dynamics through gravitational forces. Steeper slopes enhance gravity-driven transport, increasing the potential magnitude and velocity of material migration (e.g., slope runoff scouring, litter accumulation, and microbial decomposition)(Li et al., 2021). Consequently, nutrient input-output processes may differ significantly across slope gradients. Previous studies in karst peak-cluster depression areas revealed inverse relationships between slope steepness and SOC or TN contents(Zhang et al., 2013). Steeper slopes generally exhibit poorer water and nutrient retention compared to flat slopes, aligning with this study's findings: nutrient-rich values occur more frequently on flat slopes, while nutrient-poor values dominate steep and sharp slopes(Bai et al., 2013). Slope aspect governs surface processes through heterogeneous light distribution. Despite modulating factors like topography, cloud cover, and vegetation, slope aspect remains a dominant determinant of solar radiation heterogeneity. This radiation-driven heterogeneity creates distinct thermal and moisture gradients across slopes, profoundly affecting soil water-nutrient coupling processes(Li et al., 2021). Shady slopes, characterized by abundant resources and lower evaporation rates, enhance soil erosion resistance and nutrient retention, providing superior ecological niches for vegetation(Dai et al., 2023). Such habitat heterogeneity promotes optimized functional traits in shady slope vegetation and microbial communities, including higher litter production and faster nutrient turnover rates, thereby maintaining timely replenishment of soil nutrient pools(Zhang et al., 2022). In contrast, sunny slopes experience stronger light intensity, accelerating evaporation, organic matter decomposition, and nutrient loss(Geroy et al., 2011). For example, soil organic matter decomposition—a biologically mediated process—is temperature-dependent. Within certain temperature ranges, decomposition rates and magnitudes



increase exponentially(Froseth and Bleken, 2015). Elevated temperatures on sunny slopes exacerbate soil desiccation, destabilizing soil structure and aggregate stability, which intensifies nutrient depletion. This study observed frequent occurrences of nutrient-poor values on sunny or semi-sunny slopes but rarely elsewhere, consistent with the above mechanisms. At the slope scale, divergent erosion patterns and vegetation strategies drive stoichiometric heterogeneity

across slope positions. Conventional slope studies suggest that upslope areas act as "nutrient exporters," where runoff transports water and nutrients to downslope or depression zones. This results in thinner, nutrient-poor soils upslope and richer accumulations downslope(Yu et al., 2020). However, in karst regions with high bedrock exposure and fragmented topography, subsurface leakage disrupts typical runoff patterns. Some studies even report inverted nutrient gradients (e.g., higher upslope nutrient contents), likely linked to unique karst erosion processes and vegetation-mediated nutrient

dynamics(Dou et al., 2024).

In contrast to microtopographic factors, microhabitat factors in karst regions primarily drive the spatial redistribution of water and nutrients through bedrock exposure rates and the intensity of soil-rock interface exchange. Studies have shown that karst slopes, driven by the dissolution of carbonate rocks such as limestone, are often fragmented into diverse microhabitats composed of exposed bedrock and soil patches(Jiang et al., 2014). The spatial heterogeneity of bedrock

exposure critically regulates hydrological connectivity and material transport pathways, serving as a key environmental factor influencing—or even dominating—soil structure formation and the biogeochemical cycling of major nutrients(Waring et al., 2020). In stone surface microhabitats with high bedrock exposure, isolated or semi-isolated microenvironments form due to rock barriers, facilitating surface water retention and litter accumulation per unit area. This litter ultimately transforms into humus within surface crevices or depressions. Conversely, soil surface microhabitats with low bedrock exposure lack

rock barriers and exhibit flatter surfaces, weaker aggregate stability, higher soil bulk density, and greater nutrient loss due to intense leaching(Zhang et al., 2021). Additionally, carbonate rocks on karst slopes release substantial calcium ions ($Ca^{2+}$) during dissolution, with $Ca^{2+}$ concentrations significantly influencing soil organic carbon and related nutrient contents(Wang et al., 2021). In this process, stone surface microhabitats—characterized by stronger weathering and larger exchange interfaces—may develop distinct $Ca^{2+}$ release and accumulation patterns compared to other microhabitats, leading to

microhabitat-specific stoichiometric variations. Thus, under divergent hydrological pathways for nutrient input and output, the complex soil-rock interface structures across microhabitats regulate localized nutrient enrichment through dissolution-precipitation and ion exchange processes.

Soil stoichiometric characteristics in karst regions not only reflect spatial redistribution differences of nutrients across microtopographies and microhabitats but also profoundly influence plant nutrient utilization strategies and soil-vegetation

material cycling processes. While actively adapting to surface environments, vegetation participates in regulating soil stoichiometric dynamics through root activities, litter input, and decomposition(Zou et al., 2021). Studies suggest that although soil nutrient accumulation may be partially influenced by the stoichiometric traits of plant litter, the relationship between vegetation types and soil stoichiometry is not static(Bai et al., 2019). It may exhibit positive correlations in some cases(Fan et al., 2015) but no significant associations in others(Chen et al., 2022). Variance partitioning analysis (VPA) in



this study revealed low explanatory power of plant species, life forms, and nutrient content on soil stoichiometric characteristics. This may relate to karst-specific features such as fragmented topography and shallow soil layers. During sampling, we observed substantial differences in near-surface litter accumulation among plant life forms on karst slopes. Shrubs and herbs, with their ground-dwelling growth forms, effectively intercept and retain more litter for rapid decomposition. This litter includes both shrub-herb-derived material and arboreal litter transported by wind or runoff. In

contrast, arboreal species often fail to retain even their own litter around smooth trunk bases. Furthermore, studies indicate that karst soils are typically shallow. While shallow-rooted herbs thrive, most arboreal species extend roots into rock fissures to access water and nutrients. Consequently, root activities and litter-derived nutrient return in karst slope vegetation may not precisely affect their immediate soil environments. Factors such as plant species, growth stages, and community composition collectively exert direct or indirect impacts on soil stoichiometry, contributing to the complexity of plant-soil interactions in

karst slopes. Furthermore, due to the restrictive nature of the national nature reserve and the fragmentation of the karst habitat, our predefined five habitat strata (Microhabitat, Life form, Slope degree, Slope aspect, Slope position) were primarily delineated based on established research paradigms. These artificially defined discrete boundaries may fragment continuous environmental gradients. While such discretization enhances sampling feasibility, it potentially distorts the representation of microenvironment heterogeneity, constituting an inherent limitation of stratified random sampling in karst

habitat studies. Therefore, in subsequent research planned for the next year, we need to break free from this static stratification framework. We will simultaneously deploy a microenvironment sensor network to monitor light, temperature, and moisture gradients in real-time, construct continuous "factor intensity-biological response" functions, and thereby establish a machine learning-based adaptive sampling engine. This aims to achieve a paradigm shift from discrete, human-defined stratification to continuous, ecologically process-driven sampling.


## 6 Conclusions

Soil stoichiometric characteristics in karst regions exhibit synergistic adaptation to microhabitat, microtopographic, and plant life form factors, with significant correlations to specific influencing variables. The study area exhibited strong nutrient variability and pronounced spatial heterogeneity. Microhabitat factors significantly influenced soil nutrient accumulation,

forming a "rock-surface enrichment vs. soil-surface depletion" pattern. Soil surface microhabitats demonstrated nutrient impoverishment due to intense leaching, while rock surface microhabitats emerged as primary nutrient enrichment zones via unique soil-rock interface effects. However, microhabitat-driven impacts were not uniform across all elements, resulting in divergent distribution trends for certain elements and stoichiometric ratios. This underscores mechanistic differences in elemental migration and transformation. Microtopographic factors regulated soil material/energy transport and redistribution

through gravitational forces, solar radiation, and erosion intensity, fostering gradient patterns in soil stoichiometry. Yet, not all nutrients responded uniformly to these external drivers. Coupled with life form-specific nutrient regulation and uptake



strategies, these processes collectively shaped complex and heterogeneous spatial nutrient distribution patterns. Overall, soil stoichiometric characteristics in the study area are shaped by the interactive effects of microhabitat-microtopography-plant life form factors, with microhabitat-microtopography interactions dominating. Although plant life forms influenced most

elements through nutrient return and root activities, their overall impact was relatively weak and often mediated by microhabitat-microtopography conditions. These biotic-abiotic synergies reflect the inherent complexity of nutrient cycling mechanisms in karst soils.

**Data availability.** The data generated in this study are available from the first or corresponding author upon reasonable
request.

**Author contributions.** Methodology, Data curation, Writing - review & editing, Writing - original draft, Visualization, Validation: YD. Investigation, Supervision: HZ. Formal analysis, Supervision: WZ. Investigation, Validation: YC. Conceptualization, Formal analysis: CT. Formal analysis, Investigation: FD. Formal analysis, Investigation: YW.
Methodology, Supervision: RL. Funding acquisition, Visualization, Writing original draft, Writing - review & editing: PW.

**Competing interests.** The contact author has declared that none of the authors has any competing interests.

**Financial support.** This research was financially supported by the National Natural Science Foundation of China(32460275,
660 32060244).



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
