# Peer review of "Soil stoichiometric characteristics and influencing factors in karst forests under micro-topography and microhabitat scales"

_EGUsphere, 2025_

## Author Comment (AC1)

**Response to Referee 1**

Dear Editor and Referee 1,

We sincerely appreciate the opportunity to revise our manuscript (EGUSPHERE-2025-3510) and are grateful for the thorough and constructive feedback provided by the referee. Your comments have been invaluable in helping us improve the clarity, presentation, and scientific rigor of the paper. We have carefully considered all the comments and have revised the manuscript accordingly to enhance its clarity, rigor, and presentation. Below are our point-by-point responses to each suggestion.

**Comment 1**

L20: The effects of the micro-topographic parameters were investigated through correlations, but it cannot be directly stated that gravity, light exposure, and erosion were the drivers since the study didn't directly investigate those.

**Response Comment 1**

We thank the referee for this profound and constructive comment. We fully agree that, since this study did not directly measure processes such as gravity, light exposure, and erosion, it is inappropriate to directly label them as driving factors. We have revised the relevant text to more accurately state that our interpretation is based on the known indirect mechanistic roles these factors play in geomorphological and ecological contexts, rather than on direct measurements from this study. The revised text reads as follows:

Microtopographic parameters (slope degree, slope aspect, and slope position) were significantly correlated with nutrient patterns, which is consistent with their known indirect effects mediated by processes such as gravity-driven transport, differential light exposure, and erosion.

**Comment 2**

L70: The motivations for the study are presented as questions, but not listed in that format. I recommend re-wording accordingly.

**Response Comment 2**

We thank the referee for this suggestion aimed at enhancing clarity. We fully agree that reformulating the research objectives from declarative statements into explicit research questions can more directly guide the reader and establish a clear argumentative framework for the entire paper. We have revised the concluding segment of the introduction section to explicitly present the research motivations as a set of questions:

- 1. What are the spatial distribution patterns and heterogeneity of major soil nutrient contents and stoichiometric characteristics in the karst region?
- 2. How do soil nutrient elements interact with each other, and what is the intrinsic regulatory mechanism governing their stoichiometric balance?
- 3. What are the relative contributions of microhabitat types, microtopographic features, and vegetation life forms to soil stoichiometric characteristics, and how do these factors interact with each other?

**Comment 3**

L117-123: The microtopographic classes could instead be listed in a table format with the number of associated plots per class combination.

**Response Comment 3**

We thank the referee for this constructive suggestion. We agree that a table can more clearly summarize the number of sampling plots corresponding to each category of factors, including microtopography. Accordingly, we have provided a detailed summary in the Supplementary Materials (Table S1), listing the number of plots for all categorical factors (microtopography, microhabitat, and surface plant life forms). We are confident that this addition significantly enhances the readability and transparency of the sampling information, allowing the referee and readers to quickly grasp the full scope of our sampling design. We extend our gratitude once again for the insightful guidance.

Regarding the number of sampling plots across microtopographic categories, we acknowledge that while a perfectly balanced design is theoretically ideal, it was

extremely challenging to establish an equal number of plot replicates for all microtopographic factor classes due to the practical constraints of the natural karst environment in the Maolan National Nature Reserve. The distribution of microhabitats, microtopographic features, and surface plant life forms is highly heterogeneous and patchy. Our sampling strategy prioritized capturing the natural co-occurrence of these factors over artificial balance. Thus, the number of plots in each category was determined by their actual presence and distribution in the field, resulting in an unequal number of replicates. We believe this approach better represents the true structure of the ecosystem.

**Table S1.** The correspondence between various factors and the number of sample plots.

| Environmental factors | Class            | Number of Plots |
|-----------------------|------------------|-----------------|
| Slope position        | Upslope          | 6               |
|                       | Midslope         | 18              |
|                       | Downslope        | 49              |
|                       | Depression       | 13              |
| Slope degree          | Flat slope       | 16              |
|                       | Gentle slope     | 12              |
|                       | Tilted slope     | 14              |
|                       | Steep slope      | 27              |
|                       | Sharp slope      | 17              |
| Slope aspect          | Shady slope      | 6               |
|                       | Semi-shady slope | 22              |
|                       | Flat land,       | 16              |
|                       | Semi-sunny slope | 19              |
|                       | Sunny slope      | 23              |
| Microhabitat          | Stong gully      | 29              |
|                       | Stong surface    | 29              |
|                       | Soil surface     | 28              |
| Life forms            | Evergreen trees  | 32              |
|                       | Deciduous trees  | 21              |
|                       | Shrubs           | 21              |
|                       | Herbs            | 12              |

**Comment 4**

Fig. 2-5 axis labels are illegible because they are so small. Please increase text size.

**Response Comment 4**

We thank the referee for pointing out this issue. We have regenerated Figures 2 to 6, significantly increasing the font size of all axis labels and tick-mark labels to ensure that the figures remain clearly legible when scaled to the standard journal column width. The revised figures are provided below:

Figure 1. Distribution differences in stoichiometric ratios of major soil nutrients across different slope degree types, presented as violin plots overlaid with box plots. The Y-axis of each subplot denotes the values of corresponding ratios, while the X-axis represents slope degree types. An asterisk (\*) indicates significant intergroup differences (P

**Figure 2.** Distribution differences in stoichiometric ratios of major soil nutrients across different slope aspect types, presented as violin plots overlaid with box plots. The Y-axis of each subplot denotes the values of corresponding ratios, while the X-axis represents slope aspect types. An asterisk (\*) indicates significant intergroup differences (P

Figure 3. Distribution differences in stoichiometric ratios of major soil nutrients across different slope position types, presented as violin plots overlaid with box plots. The Y-axis of each subplot denotes the values of corresponding ratios, while the X-axis represents slope position types. An asterisk (\*) indicates significant intergroup differences (P

**Figure 4.** Distribution differences in stoichiometric ratios of major soil nutrients across different microhabitats, presented as violin plots overlaid with box plots. The Y-axis of each subplot denotes the values of corresponding ratios, while the X-axis represents microhabitats. An asterisk (\*) indicates significant intergroup differences (P

**Figure 5.** Distribution differences in stoichiometric ratios of major soil nutrients within the rhizosphere zones of different plant life forms, presented as violin plots overlaid with box plots. The Y-axis of each subplot denotes the values of corresponding ratios, while the X-axis represents plant life forms. An asterisk (\*) indicates significant intergroup differences (P

\* *p*

**Figure 6.** Variance partitioning analysis (VPA) of multi-factor contributions to soil stoichiometric traits. Tri-color overlapping system denotes environmental factor contributions: Yellow: Micro-environmental factors (slope/aspect/position/microhabitat); Blue: Plant structural factors (species/life form); Red: Plant nutrient factors (C/N/P contents); Values in overlapping areas indicate joint explanatory effects. Residuals=65.8%.

**Comment 10**

It seems that spaces are frequently missing between text and ellipses.

**Response Comment 10**

We thank the referee for pointing out this oversight. We have carefully reviewed the entire manuscript and corrected all similar instances to ensure that the use of spaces conforms to the required standards. We extend our sincere gratitude once again for your rigorous review.

**Comment 11**

The statistical analyses seem appropriate to the data types and questions of interest, but there is a heavy reliance on presenting descriptive statistics in the results section. Consider ways to streamline the presentation of key results, and add the

remainder to a supplemental section.

**Response Comment 11**

We thank the referee for this valuable suggestion. We fully agree that an excessive presentation of descriptive statistics in the Results section can distract readers from the core scientific findings. To enhance the clarity and focus of the results presentation, we have systematically streamlined and restructured the Results section. A substantial amount of detailed grouped descriptive statistical material has been relocated to the Supplementary Materials of the paper, with appropriate citations and references provided in the main text. The main text now retains only the most central and statistically significant findings for each factor. This approach ensures the conciseness and fluency of the main text while providing complete data support for interested readers.

**Comment 12**

Pairwise comparisons should include confidence intervals when reported in the text.

**Response Comment 12**

We thank the referee for this valuable feedback. We fully agree that reporting confidence intervals for pairwise comparisons provides more complete and transparent results.

In accordance with the referee's suggestion, we have revised all relevant sections of the manuscript reporting pairwise comparisons. The mean difference (MD), 95% confidence interval (95% CI), and P-value are now consistently provided. We believe this revision presents the statistical findings more comprehensively. Thank you again for the guidance.

**Comment 13**

The discussion is very long. While it does a good job putting the results of the distributions of plant nutrients in the context of biogeochemical conditions specific to karst regions, it is not necessary to discuss every single result in the discussion section.

They should be grouped into broader categories for a more streamlined discussion.

**Response Comment 13**

We thank the referee for their constructive comments. We fully agree that discussing each result individually can easily dilute the key points. In accordance with your suggestion, we have comprehensively restructured and streamlined the Discussion section. The key revisions are as follows:

1.We have replaced the original structure of analyzing nutrient types one by one. Instead, core findings are now categorized into key themes based on dominant environmental drivers (microhabitat, microtopography, and plant life forms). Discussions of relevant elements and their stoichiometric ratios are integrated into these thematic sections, each focusing on common patterns and underlying mechanisms.

2.We have removed repetitive background introductions and minor descriptive details. By adhering to a logical framework of "core finding - mechanistic interpretation", we now emphasize the ecological implications of soil stoichiometric characteristics, thereby more clearly revealing the inherent patterns of nutrient cycling in karst ecosystems.

This revision transforms the discussion from a descriptive listing of results into a comprehensive mechanistic analysis of nutrient heterogeneity in karst ecosystems, significantly enhancing the manuscript's depth and fluency. We thank the referee once again for their insightful review and guidance.

**Comment 14**

The broader literature referenced is usually in relation to established biogeochemical relationships, but not those specific to other karst regions. The discussion would benefit from integrating connections to studies in other karst regions globally.

**Response Comment 14**

We thank the referee for this valuable suggestion! We fully agree that comparing this study with other karst research worldwide can significantly enhance the depth and breadth of the Discussion. Accordingly, we have systematically incorporated comparisons and analyses with relevant studies from various typical karst regions—such as those in Europe and Asia—into the revised Discussion section. This revision specifically focuses on synthesizing the core conclusions and key findings from different regional studies.

By comparing the patterns observed in our results with those from other regions and conducting cross-regional integration, we have further clarified both the universality and uniqueness of our findings. This effectively expands the scope and academic depth of the Discussion, allowing our research outcomes to be better integrated into the global karst research framework. We extend our gratitude once again for the referee's attentive guidance!

**Comment 15**

L 615: Is there a citation for this statement "arboreal species often fail to retain even their own litter around smooth trunk bases" or was it only a direct observation from the present study?

**Response Comment 15**

We thank the referee for this insightful comment and apologize for the previous inaccuracy in our wording. We wish to clarify that the earlier statement—arboreal species often cannot even retain their own litter at the smooth base of the trunk—was imprecise and potentially misleading. The complete and correct explanation should read as follows:

During sampling, we observed significant differences in near-surface litter accumulation among plants of different life forms on karst slopes. Shrubs and herbs, owing to their clumped growth form close to the ground, typically intercept and retain more litter. This litter includes not only material shed by the shrubs and herbs themselves but also a substantial amount of tree-derived litter transported by wind or surface runoff. In contrast, trees, with their generally branchless trunk bases, have a weaker capacity to retain litter compared to understory shrubs and herbs growing densely near the ground. This observed

pattern is consistent with existing literature. For instance, some studies have also indicated that the understory vegetation layer can significantly intercept and regulate the spatial distribution pattern of litter—i.e., litter accumulation decreases with increasing distance from the base of understory plants. The process of litter interception by the understory vegetation may alter the microenvironment (including light, moisture, soil, and microbial communities), thereby influencing its decomposition trajectory.

**Comment 16**

L625-629: It's great to have this plan, but it is more of a statement for a research proposal than an article. Perhaps something like "Future research directions should explore more dynamic approaches to characterizing karst microenvironments, potentially incorporating real-time environmental monitoring, continuous gradient analysis, and adaptive sampling strategies driven by ecological processes rather than predetermined spatial categories. This could enable better representation of the continuous nature of environmental variation in heterogeneous karst systems."

**Response Comment 16**

We thank the referee for this highly pertinent and constructive comment. We fully agree that the original description of future work read more like a research plan than appropriate content for the Conclusions section of an academic paper. Following the referee's suggestion, we have completely rewritten this passage, reframing it from a "plan for future work" into an "outlook for future research directions". The revised text reads as follows:

Therefore, future research should transcend this static stratification framework and commit to adopting continuous environmental gradient monitoring and high-resolution sampling strategies to overcome this limitation. This will better capture the complexity of karst ecosystems and facilitate a paradigm shift from discrete stratification to process-driven approaches.

---

## Author Comment (AC2)

**Response to Referee 2**

Dear Editor and Referee 2,

We sincerely appreciate the opportunity to revise our manuscript (EGUSPHERE-2025-3510) and are grateful for the thorough and constructive feedback provided by the referee. Your comments have been invaluable in helping us improve the clarity, presentation, and scientific rigor of the paper. We have carefully considered all the comments and have revised the manuscript accordingly to enhance its clarity, rigor, and presentation. Below are our point-by-point responses to each suggestion.

**Major Comment 1**

Consistence of terms needs to be improved. A few examples:

Types of microhabitat were rock surface, rock groove, and soil surface (lines 125-144), but they were coded as stone surface, stong gully and soil surface from the method (lines 182) and used in the results and discussion. I believe 'stong' is a typo, but it appeared eight times in the texts and sometimes is capitalized and sometimes not.

Hyphen usage: e.g. microtopography and micro-topography both appeared in the text. Microenvironment, micro-environment, etc.

Slope degrees and slop gradients: stick to one term.

**Response Major Comment 1**

We sincerely appreciate the referee's meticulous review and valuable suggestions. The issues raised regarding terminology inconsistency and typographical errors are highly important, and we fully accept them. We apologize for these oversights, which indeed affected the manuscript's rigor and readability. Following your comments, we have conducted a thorough check and made unified revisions throughout the text. The specific modifications are detailed below:

(1) We have corrected all typographical errors and standardized the relevant terminology. Specifically, microhabitat terms have been unified as: "stone gully," "stone surface," and "soil surface." Slope description terminology has been

standardized as "slope degrees." Incorrect expressions in the main text, such as "rock groove," "rock surface," "stong gully," and "slop gradients," have been rectified. Relevant proper nouns in the main text are no longer capitalized unless they appear at the beginning of a sentence.

(2) We have adopted the non-hyphenated format as the standard to maintain conciseness and align with common practices in modern academic writing. Consequently, terms such as "micro-topography" and "micro-habitat" have been uniformly revised to "microtopography" and "microhabitat." Other similar terms have also been standardized without hyphens.

**Major Comment 2**

Missing important information, for example:

Lines 100-108: how many sites/plots were sampled and how many points were in each stratum? The authors neither report this in any tables nor figures.

**Response Major Comment 2**

In Table S1, we detail the number of plots corresponding to each hierarchical category under microtopography (slope position, slope degree, slope aspect), microhabitat, and aboveground plant life forms. Three sampling points were established in each plot.

Table S1. The correspondence between various factors and the number of sample plots

| Environmental factors | Class            | Number of Plots |
|-----------------------|------------------|-----------------|
| Slope position        | Upslope          | 6               |
|                       | Midslope         | 18              |
|                       | Downslope        | 49              |
|                       | Depression       | 13              |
| Slope degree          | Flat slope       | 16              |
|                       | Gentle slope     | 12              |
|                       | Tilted slope     | 14              |
|                       | Steep slope      | 27              |
|                       | Sharp slope      | 17              |
|                       | Shady slope      | 6               |
| Slope aspect          | Semi-shady slope | 22              |
|                       | Flat land,       | 16              |
|                       | Semi-sunny slope | 19              |
|                       | Sunny slope      | 23              |
|                       | Stone gully      | 29              |
| Microhabitat          | Stone surface    | 29              |
|                       | Soil surface     | 28              |
|                       | Evergreen trees  | 32              |
| 1.0.0                 | Deciduous trees  | 21              |
| Life forms            | Shrubs           | 21              |
|                       | Herbs            | 12              |

It should be noted that although a perfectly balanced sample design is theoretically more ideal, it was exceptionally challenging to achieve equal replication across all factor categories in the natural karst environment of the Maolan National Nature Reserve. This difficulty stems from two primary reasons: Firstly, the area is characterized by typical karst peak-cluster depression topography, which is highly fragmented and significantly differs from the continuous and uniform slopes of non-karst regions. Secondly, as a natural forest ecosystem, the distribution and combination of vegetation exhibit a high degree of natural randomness, unlike the regular and homogeneous patterns typical of plantations. Consequently, not all theoretical combinations of factors exist in reality. Under this practical constraint, the sampling strategy of this study prioritized accurately reflecting the natural co-occurrence relationships and spatial heterogeneity of environmental factors over

pursuing an artificially set balanced sample size. The number of plots per category was solely determined by their actual presence and distribution in the field, resulting in varying numbers of replicates. We believe this strategy more faithfully represents the true structure and habitat complexity of the karst natural forest ecosystem.

We believe that the supplementary information and explanations provided above sufficiently clarify the distribution of sample sizes. Although the sample sizes are unequal, the number of samples under each classification level meets the basic requirements for robust statistical analysis and is adequate to support the relevant conclusions of this study.

**Major Comment 3**

Lines 145-149: The description of microhabitat showed soil layer thickness was less than 20 cm for rock surface, <20 cm for rock groove and has well-developed soil horizons for soil surface type, what was the sampling depth and how did the authors sample soils while thickness differs this much? Sampled the center of the layer with cut ring? Was the thickness of soil layer recorded? The comparability of analyzed properties and calculated stoichiometric ratios would be questionable if the sampling depths were not similar.

**Response Major Comment 3**

The soil surface microhabitat is characterized by relatively continuous soil cover or a low bedrock exposure rate. As the study area is a typical karst region, the surface soil layer is generally extremely shallow. During field investigations, we observed that soil thickness in most areas, including the soil surface microhabitat, is less than 20 cm. Therefore, although the soil surface microhabitat has relatively continuous and uniform soil cover, this does not imply that its soil thickness is significantly greater than that of stone surface and stone gully microhabitats.

Given the extremely limited soil thickness across the entire study area, soil sampling could not be stratified during collection. Sampling was completed using only a shovel and a cutting ring (100 cm3). Upon arriving at a sampling point, the sampler selected the root distribution area of the plant (within a 20 cm radius) as the

sampling range. After clearing surface litter, the average soil depth at the sampling point was measured with a steel ruler. Then, three undisturbed soil cores were collected from the middle layer of the soil using the cutting ring. After manually removing non-soil materials, the samples were mixed thoroughly, ensuring each sample had a net weight of  $\geq 500$  g. After collection, the samples were immediately coded, packaged in sterile sampling bags, and temporarily stored in a portable cooler for subsequent laboratory analysis.

**Specific comments**

Comment 1

Line 18: specify the microhabitat factors

Response:

We thank the referee for this valuable suggestion. We fully agree that specifying

the "microhabitat factors" in the abstract makes the presentation of the research

findings clearer. Following the suggestion, we have revised the relevant sentences

near line 18 of the abstract as follows:

(2) Microhabitat factors (stone gully, stone surface, soil surface) significantly

influenced nutrient accumulation, though different elements showed distinct

response patterns to microhabitat variations;

Comment 2

Lines 21-23: different instead of differential?

**Response Comment 2**

We thank the referee for the suggestion regarding terminology. We agree with

your perspective and have revised the original text by replacing "differential" with

"different" to enhance clarity and precision. The modified sentence reads as follows:

Different response mechanisms of nutrients to microtopographic and

microhabitat factors, combined with the different nutrient regulation and

absorption strategies of various plant life forms, collectively shaped the complex

stoichiometric characteristics.

This revision was made based on the following considerations: "different" more

directly emphasizes the distinctions between nutrient response mechanisms and plant

strategies. In the context of ecological stoichiometry research, "different" is a clearer

and more unambiguous term for describing "distinct mechanisms/strategies," which

improves readability without compromising scientific rigor. All core terminology

remains consistent with the term system used throughout the manuscript.

**Comment 3**

Line 22: specify abiotic factors to microtopographic and microhabitat instead of using abiotic factors alone. 'Abiotic factors' here may mislead readers to think about temperature and other commonly used climatic factors.

**Response Comment 3**

We thank the referee for this crucial revision suggestion. We fully agree that the use of the broad term "abiotic factors" in the abstract was indeed imprecise and could have misled readers into considering other commonly referenced climate factors such as temperature. Following the referee's valuable advice, we have specified these as "microtopographic and microhabitat factors" in the abstract to improve accuracy and clarity. The revised text is as follows:

Differential response mechanisms of nutrients to microtopographic and microhabitat factors, combined with the differential nutrient regulation and absorption strategies of various plant life forms, collectively shaped the complex stoichiometric characteristics.

This modification sharpens the focus of our study, accurately confining the conclusions to the specific microenvironmental drivers examined—such as slope degree, slope aspect, slope position, and microhabitats including stone gullies, stone surfaces, and soil surfaces—thus effectively avoiding potential confusion with macroclimatic factors. We once again express our gratitude for your insightful review, which has significantly enhanced the rigor of our manuscript's presentation.

**Comment 4**

Lines 70-75: rephrase them into questions or aims?

**Response Comment 4**

We thank the referee for this highly valuable suggestion. We fully agree that reframing the key research points as explicit research questions/objectives can more clearly outline the scientific intent and logical framework of this study. Following the referee's advice, we have comprehensively revised the originally declarative key points in the Introduction section into a set of specific research questions. The

modifications are detailed below:

- 1. What are the spatial distribution patterns and heterogeneity of major soil nutrient contents and stoichiometric characteristics in the karst area?
- 2. What are the interrelationships among soil nutrient elements, and what is the intrinsic regulatory mechanism governing their stoichiometric balance?
- 3. What are the relative contributions of microhabitat types, microtopographic features, and vegetation life forms to soil stoichiometric characteristics, and how do these factors interact with each other?

**Comment 5**

Line 76: what multi-scale refers to? Is not this study at the hillslope scale?

**Response Comment 5**

We thank the referee for this important comment. We agree that the present study essentially constitutes an integrated analysis of different influencing factors conducted primarily at the hillslope scale. The use of the term "multi-scale" was indeed imprecise and potentially misleading. Following your suggestion, we have removed the expression "multi-scale" from the manuscript and replaced it with "multi-factor integrated analysis" to more accurately reflect the research content. Please see below for the specific revision:

This study employs a multi-factor integrated analysis to elucidate the formation mechanisms of karst soil stoichiometry, providing novel theoretical frameworks for understanding soil-vegetation co-adaptation mechanisms in karst ecosystems, while offering scientific underpinnings for ecological restoration and sustainable management practices in this region.

**Comment 6**

It looks like space is often missing before each citation.

**Response Comment 6**

We thank the referee for their valuable feedback on the formatting details of our manuscript. We fully agree that the issue of missing spaces before citation markers is an important aspect of ensuring standardization and professionalism in academic writing.

In accordance with your suggestion, we have systematically checked the entire manuscript and added standard spaces between the main text and citation markers to ensure full compliance with academic formatting requirements.

**Comment 7**

Line 96: section 2 study area and 3 study method can be combined into one.

**Response Comment 7**

We thank the referee for their valuable suggestions regarding the structure of the manuscript. We fully agree that merging the "Study Area" and "Research Methods" sections into a single chapter can more clearly demonstrate how the characteristics of the study area directly influence the methodology, thereby enhancing the logical flow and structural coherence of the paper.

Following your recommendation, we have combined the original "2 Study area" and "3 Research methods" sections into a new chapter titled: "2 Study area and methods" Within this new chapter, we sequentially describe the physical geography of the study area, the plot establishment, the criteria for classifying microtopographic and microhabitat factors, the procedures for sample collection and processing, and the methods for statistical data analysis.

**Comment 8**

Lines 155-158: 0.149 mm sieve, the justification here is not convincing. Up to 2 mm is still considered as soil in commonly used soil particle size classification systems. What happened to the material not passing 0.149 mm sieve? Dropped or milled until passing the sieve? Add details about which properties were analyzed with fresh soil and which with dry soils.

**Response Comment 8**

We thank the referee for their valuable comments regarding the methodological details. In response to your suggestions for revising the "Sample Processing and

Determination" section, we provide the following clarifications:

1.Rationale for selecting the 0.149 mm sieve: The analysis of total soil elements (e.g., total nitrogen, total phosphorus, total potassium) and soil organic carbon requires samples to be ground to a specific fineness. This ensures complete and homogeneous decomposition during subsequent digestion or fusion, thereby eliminating analytical errors caused by particle size heterogeneity and guaranteeing the accuracy of total nutrient measurements. Coarser particles (<2 mm but >0.149 mm) may contain encapsulated minerals that react incompletely during digestion, leading to underestimated "total" element results. Grinding to 100-mesh significantly increases the reactive surface area, ensuring complete digestion, which is a critical step for obtaining accurate data. Consequently, the 0.149 mm (100-mesh) size is recommended by many international standard soil analysis methods.

2.Treatment of material not passing through the sieve: All residual material not passing the 0.149 mm sieve was subjected to secondary grinding using an agate mortar until it completely passed through. This portion was then thoroughly mixed with the sieved fine soil fraction to form an analytical sample representing the entirety of the original soil matrix (after removal of >2 mm gravel). This procedure ensures that elements from all soil components are included, preventing deviation from true values and safeguarding data accuracy and representativeness.

3. Sample state: All indicators in this study were determined using air-dried soil samples, following the *Chinese Forestry Industry Standards (LY/T 1210 ~1275-1999)*. We ensured the timeliness and consistency of the air-drying process and stored samples in a cool, dry place to minimize any potential effects attributable to air-drying. Pretreatment and analytical conditions were consistent for all samples to ensure data comparability and reliability.

Your comments have significantly enhanced the rigor and clarity of our methodological description. We have integrated all the above details into the revised manuscript and reiterate our sincere appreciation for your thorough guidance.

**Comment 9**

Lines 172-173: explicitly list which ratios were calculated.

**Response Comment 9**

We thank the referee for the valuable suggestion. Clearly listing the calculated stoichiometric ratios indeed significantly enhances the clarity of the Methods section and the reproducibility of the study. Following your advice, we have revised the original text in the "2.5 Data processing and analysis" section to explicitly list all the calculated stoichiometric ratios. The specific modifications are detailed below:

This study used mass contents to characterize soil nutrient indicators (SOC, TN, TP, etc., totaling 11 items; see Supplementary Table S2 for details). The stoichiometric ratios of the elements were calculated as mass ratios (SOC:TN, SOC:TP, SOC:TK, etc., totaling 9 items; see Supplementary Table S3 for details).

This revision ensures that readers can clearly understand all the variables we analyzed, which correspond fully to the data presented in the Results section. We thank the referee again for helping us improve the rigor of the manuscript.

**Comment 10**

Table 1 caption, it is not a comparison, use soil properties instead of detection indicators. Is CV the coefficient variation? Add reference for which range is considered as weak, moderate and strong variability.

**Response Comment 10**

We thank the referee for their valuable suggestions regarding the table terminology and data analysis details. We have implemented the following revisions in accordance with your recommendations to enhance the manuscript's rigor:

- 1. Regarding the title of Table 1: We have revised the title to "Analytical methods and core instrumentation for determining soil properties." This modification more accurately represents the table's nature as a methodological inventory and adopts the more standardized terminology "soil properties" within the field.
  - 2. Regarding the definition and classification criteria for CV: In Section "2.5

Data processing and analysis," we have explicitly defined "CV" as the abbreviation for "coefficient of variation" and supplemented the variability intensity classification criteria (CV  $\leq$  0.20 for weak variability, 0.20

Figure 1. Distribution differences in stoichiometric ratios of major soil nutrients across different slope degree types, presented as violin plots overlaid with box plots. The Y-axis of each subplot denotes the values of corresponding ratios, while the X-axis represents slope degree types. An asterisk (\*) indicates significant intergroup differences (P < 0.05), with black horizontal lines connecting groups exhibiting differences. The number of samples (n) for each slope degree type is as follows: flat slope (n=16), gentle slope (n=12), tilted slope (n=14), steep slope (n=27), sharp slope (n=17).

We believe that with the addition of sample size information, the data foundation presented in the figures is now clearer, further strengthening the rigor of the manuscript. We sincerely appreciate your meticulous and professional review.

**Comment 14**

Lines 229-231: wordy, can be reduced to e.g. Major soil nutrients were highest at flat land, followed by shady slopes and sunny slopes. There are many places like this

can be shortened, consider using editing services or asking native speakers to improve language.

**Response Comment 14**

We thank the referee for their valuable comments on the language expression in our manuscript. We fully agree that the original wording was not concise enough and may have obscured the emphasis on key findings. Following your suggestion, we have revised the sentence into a more succinct expression: "Major soil nutrients were highest on flat land, followed by shady slopes, and lowest on sunny slopes." This revision removes redundant words and improves clarity. Furthermore, we have conducted a thorough check of the entire manuscript and performed similar simplification of other verbose expressions to optimize linguistic fluency and ensure compliance with academic writing standards. We believe these revisions significantly enhance the readability and academic value of the paper. Thank you again for your helpful review.

**Comment 15**

Lines 321-325: this is because the same variable was used in correlating that ratio, it does not make sense to report this type of correlation. Report only the most relevant parts.

**Response Comment 15**

We thank the referee for emphasizing this crucial point. We fully understand and agree that reporting mathematical autocorrelation caused by the same variable lacks scientific significance and may lead to misleading findings. Following your guidance, we have thoroughly rewritten the correlation analysis section, focusing exclusively on reporting the most relevant and ecologically meaningful relationships between different variables (e.g., between distinct elements or between different ratios). The specific revisions are as follows:

The correlation matrix revealed complex interrelationships among soil nutrients and their stoichiometric characteristics (Fig. 7). Soil SOC and TN contents exhibited a strong co-variation trend (r = 0.94, P < 0.01), and both

showed significant positive correlations with most other nutrients (e.g., HN, TCa, and ExCa). In contrast, TK content was significantly negatively correlated with SOC, TN, and several key stoichiometric ratios (e.g., SOC:TP, TN:TMg). Available phosphorus (AP) demonstrated a more independent pattern, showing significant positive correlations only with TCa and SOC.

Close associations were also observed among different stoichiometric ratios; for instance, SOC:TK and TN:TK exhibited a highly significant positive correlation (r = 0.95, P < 0.01). These association patterns indicate tightly coupled relationships among major nutrient elements such as carbon, nitrogen, and calcium, as well as the unique distribution pattern of certain individual elements within the karst soil system. The underlying driving mechanisms will be thoroughly analyzed in the Discussion section.

**Comment 16**

I think section 4.4 does not need subsections, readers can see that one paragraph reporting results from RDA and the other from VPA.

**Response Comment 16**

We thank the referee for their valuable suggestions regarding the manuscript's structure. We agree that presenting the RDA and VPA analyses in separate subsections would create unnecessary redundancy. As recommended, we have integrated the relevant content into a single, coherent section. The revised narrative now follows a more logical progression: it first identifies the key environmental drivers through RDA and correlation analysis, then quantifies their individual and joint contributions via VPA, thereby systematically revealing the influence of each factor. We believe this integration yields a more concise and fluid structure, strengthens the logical flow of the argument, and enhances readability. We are grateful for your insightful comments, which have improved our manuscript.

**Comment 17**

Figure 8: space is enough to write environmental factors in full in the biplot, or

add the abbreviations to the figure caption.

**Response Comment 17**

Thank you for your valuable comment regarding the clarity of Figure 8(now Figure 7). We agree that it is essential for readers to understand the meaning of each environmental factor abbreviation in the biplot. We have followed your suggestion and added a key explaining the abbreviations and full names of the environmental variables in the figure caption. The specific revision is as follows:

Figure 7. Ordination biplot of redundancy analysis (RDA) for soil stoichiometric traits and environmental factors. Axes: RDA1 (16.3% variance explained) and RDA2 (6.8%). Blue arrows: stoichiometric variables. Red arrows represent environmental variables: MH (Microhabitat), SP (Slope Position), SA (Slope Aspect), SD (Slope Degree), LF (Life Form). Arrow length denotes variable contribution; inter-arrow angles reflect correlations. Origin (0,0) serves as the reference point.

This modification ensures that all information in the figure is directly accessible to the reader without compromising the clarity of the plot. We believe the revised figure now fully addresses your concern. Thank you again for your thorough review.

**Comment 18**

Line 336, and table 4: what type of correlation is used in correlating continuous variables and assigned ordinal variables? What is the correlation coefficient called in this case? I could not find details from the method either. It seems that results from Table 4 is not used from the results section, move to supplement instead?

**Response Comment 18**

We thank the referee for their insightful comments regarding the statistical methods and results presentation.

1. Regarding the correlation analysis method: You are absolutely correct. We have explicitly stated in Section "2.7 Data processing and analysis" that Spearman's rank correlation analysis was uniformly employed in this study to assess the relationships between soil stoichiometric characteristics (continuous variables) and

numerically coded environmental factors (ordinal variables). We sincerely apologize for the omission of this key methodological detail in the initial draft and thank you for highlighting it.

2. Regarding Table 4: Following your suggestion, we have moved the detailed Spearman's correlation coefficient table from the main text to the Supplementary Materials, renumbering it as Table S9. The corresponding table caption has been revised to: "Table S9. Spearman's rank correlation coefficients between soil stoichiometric characteristics and environmental factors". In the main text, we no longer list the data but instead summarize the key findings, integrating them with the RDA and VPA results to create a more focused and coherent discussion.

We believe this adjustment significantly enhances the conciseness and readability of the Results section. We are truly grateful for your perceptive suggestions.cantly enhance the conciseness and readability of the Results section. Thank you once again for your insightful suggestions.

**Comment 19**

Line 359-360: plant species and plant nutrient contents appeared first time here, describe them in the method.

**Response Comment 19**

We thank the referee for the valuable comments regarding the methodological details. The observation concerning the lack of description for plant species identification and plant nutrient analysis methods is highly pertinent. Accordingly, we have addressed this by adding a new subsection, "2.6 Vegetation survey and plant nutrient analysis", within the "2 Study area and methods". The specific additions are as follows:

During soil sample collection, field personnel simultaneously recorded the plant species and life forms at each sampling point and collected representative leaf samples. Plant species were identified to the species level using taxonomic methods, and their Latin names were documented. Plant life forms were classified into four categories—evergreen trees, deciduous trees, shrubs, and

herbs—based on standard botanical criteria and adapted to the local conditions of the study area. Leaf sampling followed the principle of representativeness: using high-pruners, well-developed branches from the east, south, west, north, upper, middle, and lower parts of the canopy were clipped. Fully expanded, disease-free, intact leaves without petioles were then picked from these branches. The collected leaves were thoroughly mixed, and a subsample of 30–50 leaves was retained using the quartering method. These samples were labeled, sealed in zip-lock bags, and stored in a portable refrigerator for subsequent analysis.

The preparation of plant samples referred to the industry standard LY/T 1267–1999. Carbon (C) content was determined by the potassium dichromate oxidation–external heating method; nitrogen (N) content by the Kjeldahl method; phosphorus (P) content by the molybdenum–antimony anti-spectrophotometric method; potassium (K) content by flame photometry; and calcium (Ca) and magnesium (Mg) contents by atomic absorption spectrophotometry (Table 2). All analytical procedures were strictly conducted in accordance with the *Chinese Forestry Industry Standards (LY/T 1210–1275-1999)*.

Table 1 Analytical methods and core instrumentation for determining plant properties

| Detection
Indicator | Standard Method                                          | Core Instrumentation                                                    |
|------------------------|----------------------------------------------------------|-------------------------------------------------------------------------|
| С                      | Potassium dichromate oxidation - external heating method | Oil bath (180 $^{\circ}$ C $\pm$ 0.5 $^{\circ}$ C), Titration apparatus |
| N                      | Kjeldahl method                                          | Kjeldahl nitrogen apparatus, Digestion furnace                          |
| P                      | Molybdenum - antimony anti-spectrophotometric method     | Muffle furnace, Spectrophotometer                                       |
| K                      | Flame photometry                                         | Muffle furnace, Flame photometer                                        |
| Ca & Mg                | Atomic absorption spectrophotometry                      | Atomic absorption spectrometer (AAS)                                    |

This supplement ensures the completeness of the methodological description and provides a clear data source and experimental basis for the plant-related factors mentioned in the Results section. We thank the referee once again for the rigorous

review, which has significantly enhanced the scientific soundness of the manuscript.

**Comment 20**

Lines 360-367 and figure 9: What are stoichiometric traits and stoichiometric variation refer to? What is/are response variable(s) in VPA analysis?

**Response Comment 20**

We thank the referee for their valuable comments regarding the clarity of terminology. We apologize for the insufficiently clear definitions of key terms such as "stoichiometric traits" and "stoichiometric variation" in the original manuscript. Your feedback has significantly enhanced the rigor of our paper. Following your suggestions, we have unified and refined the relevant expressions throughout the text, particularly in the Methods and Results sections.

- 1. In this study, the use of the term "stoichiometric traits" was inaccurate. Our intended meaning specifically refers to all the concrete soil stoichiometric indicators we measured. These comprise two main categories:
- (1) **Soil nutrient contents**: including the contents of 11 specific elements such as SOC, TN, and TP.
- (2) **Soil stoichiometric ratios**: referring to the 9 key ratios calculated from the aforementioned nutrient contents, such as SOC:TN and SOC:TP.

In the VPA statistical analysis, these specific indicators collectively form the response variable matrix. Consequently, we have revised the originally used, imprecise expressions like "soil stoichiometric traits" to "soil stoichiometric characteristics," which accurately denotes this measurable, multivariate dataset.

2. In this study, the use of the term "stoichiometric variation" was inaccurate. Our intended meaning specifically refers to the **total variance** of the multivariate data matrix composed of all response variables used in the VPA analysis (i.e., the 11 soil nutrient contents and the 9 stoichiometric ratios). The purpose of the VPA is to partition this total 'variation' (100%) into parts explainable by different environmental factor groups (microenvironment, plant structure, plant nutrients) and the residual, unexplained portion. Therefore, we have corrected imprecise statements in the

original text, such as "The results collectively explained 34.21% of soil stoichiometric variations," to: "The results collectively explained 34.21% of the total variance in the soil stoichiometric dataset."

These revisions thoroughly eliminate terminological ambiguity and directly address your question regarding "what the response variables are." We extend our sincere gratitude once again for your insightful review.

**Comment 21**

Regarding the discussion part, the authors tended to begin paragraphs by stating the importance of the topics, this fits to introduction better and completely unnecessary in the discussion. I suggest the authors focus on their findings and then put them into a larger context (findings from other studies). As an example, take a look at https://doi.org/10.5194/bg-19-2171-2022. In addition, sometimes it is not clear whether the authors referring to their results or the results from other studies in the discussion, the clarity could be improved.

**Response Comment 21**

We thank the referee for the profound and constructive comments regarding the Discussion section of our manuscript. The issues raised accurately identified key shortcomings in our academic expression and focus, which we fully acknowledge. We have completely restructured and rewritten the Discussion section to thoroughly address the concerns regarding logical flow and clarity of presentation. During revision, we carefully studied and drew inspiration from the exemplary paper recommended by the referee (Spohn & Stendahl, 2022, *Biogeosciences*), reshaping our narrative approach accordingly. The specific revisions are detailed below:

1. Addressing the comment that "paragraphs should begin by focusing on research findings rather than the general importance of the topic":

We recognized that opening multiple paragraphs with general background knowledge indeed weakened the focus of the discussion. In response, we fundamentally restructured the opening sentences of all paragraphs to ensure each section starts directly and explicitly with the specific findings of this study.

Statements of general knowledge, such as "Carbon is the most common element in plants," were removed. This ensures that every part of the revised discussion revolves squarely around "what we found" and "what this finding means," thereby shifting the focus entirely to the interpretation of our own results.

This approach immediately engages the reader with the core findings of our study before contextualizing them within the broader literature, strictly adhering to the principle of "findings first, context second."

2. Addressing the need to "situate findings within a broader context (other studies)":

Following the presentation of each key finding, we actively connected and compared it with existing research. We paid particular attention to supplementing and citing studies from other karst ecosystems, moving beyond reliance solely on fundamental biochemical theories or local studies. This strengthens the argument regarding the universality and specificity of our findings, significantly enhancing the academic depth and breadth of the discussion.

3. Addressing the comment regarding "unclear distinction between cited literature and own results":

We acknowledged that ambiguous referencing weakened the clarity and credibility of our arguments. Consequently, we implemented strict distinctions throughout the Discussion section as follows:

- (1) When stating our own results, we explicitly used phrases such as "In this study, we found that..." and "Our data revealed that...".
- (2) When citing others' work for comparison or support, we clearly used constructions like "This is consistent with the findings of [Author] in [Region]..." and "In contrast to the results of [study]...".

This rigorous terminological distinction makes it immediately clear which elements are our novel contributions and which are references to or comparisons with existing knowledge, completely avoiding any potential confusion.

We believe that through these meticulous and substantial revisions, the Discussion section now more clearly, powerfully, and appropriately articulates the

value of our research in accordance with academic standards. We reiterate our sincere gratitude for the valuable time and guidance provided, which has been instrumental in enhancing the quality of our manuscript.

**Comment 22**

Lines 620-630: The limitation and outlook could be in a separate paragraph. No need to list detailed next steps, instead, point out the direction is sufficient.

**Response Comment 22**

We thank the referee for this highly pertinent and constructive comment. Following your suggestion, we have consolidated the discussion on limitations and future outlook into a distinct paragraph within the Discussion section. The revised text now reads:

It should be noted that the sampling strategy of this study was based on predefined discrete habitat stratifications. While this approach enhanced sampling feasibility within the complex karst terrain, such discretization may not fully capture continuous environmental gradients, thereby constituting an inherent limitation in characterizing microenvironmental heterogeneity and sampling design in such habitats. Therefore, future research should transcend this static stratification framework and commit to adopting continuous environmental gradient monitoring and high-resolution sampling strategies to overcome this limitation. This will better capture the complexity of karst ecosystems and facilitate a paradigm shift from discrete stratification to process-driven approaches.

We believe this revision brings clearer structure to the Discussion and effectively addresses your comment. Thank you once again for your insightful suggestion.